# Intermittent antibiotic treatment of bacterial biofilms favors the rapid evolution of resistance

Masaru Usui [1,2] ✉, Yutaka Yoshii[2], Stanislas Thiriet-Rupert[2], Jean-Marc Ghigo [2] & Christophe Beloin [2] ✉

Bacterial antibiotic resistance is a global health concern of increasing importance and intensive study. Although biofilms are a common source of infections in clinical settings, little is known about the development of antibiotic resistance within biofilms. Here, we use experimental evolution to compare selection of resistance mutations in planktonic and biofilm *Escherichia coli* populations exposed to clinically relevant cycles of lethal treatment with the aminoglycoside amikacin. Consistently, mutations in *sbmA*, encoding an inner membrane peptide transporter, and *fusA*, encoding the essential elongation factor G, are rapidly selected in biofilms, but not in planktonic cells. This is due to a combination of enhanced mutation rate, increased adhesion capacity and protective biofilm-associated tolerance. These results show that the biofilm environment favors rapid evolution of resistance and provide new insights into the dynamic evolution of antibiotic resistance in biofilms.

[1] Laboratory of Food Microbiology and Food Safety, Department of Health and Environmental Sciences, School of Veterinary Medicine, Rakuno Gakuen University, Hokkaido, Japan. [2] Institut Pasteur, Université de Paris Cité, UMR CNRS 6047, Genetics of Biofilms Laboratory, 75015 Paris, France. ✉email: usuima@rakuno.ac.jp; christophe.beloin@pasteur.fr

Bacterial infections are a leading cause of morbidity and mortality commonly treated with antibiotics. However, increased use of antibiotic treatments has led to the propagation of antibiotic resistance genes by horizontal gene transfer or the selection of vertically transmitted mutations. This global increase in antibiotic resistant bacteria, particularly in pathogenic strains resistant to treatment, is a major health concern[1,2]. Adaptive laboratory evolution experiments have been used to reproduce the dynamics of emergence and selection of antibiotic resistance in various bacteria. Exposing planktonic bacteria to sub-inhibitory or progressively increasing concentrations of antibiotics quickly leads to diverse inheritable resistance mutations[3]. By contrast, the use of periodic and short (3–8 h) lethal antibiotic treatments leads to mutants that have increased tolerance to antibiotics, i.e., their ability to survive but not grow under antibiotic pressure, for instance due to increased lag time or reduction of proton motive force[4–10]. In some cases, this increased tolerance favoured the emergence of genetic resistance[11–14].

Although antibiotic tolerance is relatively understudied compared to genetic resistance[15–17], it is the hallmark of surface-attached bacterial communities called biofilms[18,19]. Biofilms indeed display a characteristic high level of tolerance to a broad range of antibiotics that disappears quickly after biofilm dispersion. Consequently, even when caused by non-resistant bacteria, biofilm-associated infections are difficult to eradicate and regrowth of surviving biofilm bacteria when antibiotic treatment stops is a typical cause of therapeutic failure due to bacterial infection relapse[19–21]. The emergence of antibiotic resistance within a tolerant biofilm population could therefore constitute an aggravating factor increasing the frequency of therapeutic failure and infection recurrence.

Whereas the horizontal transfer of antibiotic resistance genes is well-established in biofilms[22–28], the emergence and evolution of genetically and vertically inheritable antibiotic resistance in biofilms is less understood. Physico-chemical heterogeneity and diffusion limitation in biofilm environments may reduce exposure to antibiotics[29], induce bacterial stress response and increase mutation rates compared to planktonic conditions, leading to genetic diversification[30–35]. Consistently, *Escherichia coli* biofilms rapidly acquire resistance to several different classes of antibiotics, even in the absence of antibiotics[36–38]. *Enterococcus faecalis* biofilms formed in continuous flow conditions and exposed to increasing concentrations of daptomycin were also shown to evolve resistance mutations correlating with the acquisition of increased biofilm formation capacities[39]. Likewise, *Acinetobacter baumannii* biofilms treated for 3 days with sub-minimal inhibitory concentration (MIC) of antibiotics displayed higher survival through enhanced biofilm formation (for tetracycline) or by increasing drug resistance (for ciprofloxacin)[40].

Although biofilm-associated antibiotic tolerance combined with increased genetic diversity is of growing concern, comparison of antibiotic resistance evolution in planktonic and biofilm populations has led to contradictory results, and no clear evolutionary rate difference has been observed between the two lifestyles. On one hand, *Pseudomonas aeruginosa* colony biofilms continuously subjected to sub-MIC of ciprofloxacin evolved towards resistance faster than planktonic bacteria[41]. By contrast, *A. baumannii* biofilms formed on beads and subjected to similar conditions or stepwise increased concentration of ciprofloxacin evolved resistance slower than planktonic populations[42]. No differences were observed between biofilm and planktonic population evolution rates in studies of *Salmonella* Typhimurium bead biofilms continuously treated with sub-inhibitory concentrations of azithromycin, cefotaxime, and ciprofloxacin[43] or for *A. baumannii* and *P. aeruginosa* bead biofilms treated with stepwise increased concentration of tobramycin[44]. Moreover, whereas in some of these experiments, higher MICs were reached in evolved planktonic conditions[41,42,45], in others, no differences in MIC could be observed between planktonic and biofilm evolved bacteria[44]. Hence, it is still unclear whether, compared to planktonic bacteria, antibiotic-tolerant biofilms constitute a more dynamic evolutionary reservoir enabling the emergence of antibiotic resistance.

Here, we performed adaptive evolution experiments to investigate the evolutionary paths toward increased antibiotic resistance in pathogenic *E. coli* planktonic and biofilm populations subjected or not to antibiotic pressure. Considering that the management of chronic biofilm-associated infections often requires the repeated administration of high concentrations of antibiotic[19,21,46–48], we exposed *E. coli* planktonic and biofilm populations to long and intermittent 24 h treatment with lethal concentrations of the aminoglycoside amikacin over 10 cycles of evolution. Amikacin is a bactericidal antibiotic that binds to bacterial ribosomes, inhibits protein synthesis and causes membrane disruption through production of aberrant proteins[49]. The use of this antibiotic treatment protocol enabled us to show that biofilms, unlike planktonic populations, exhibit a rapid evolution towards increased MIC to amikacin. By contrast with what was observed with shorter periodic treatments of planktonic populations, neither biofilm nor planktonic bacteria accumulated mutations previously associated with tolerance to aminoglycoside (i.e., in *nuoN*, *oppB* or *gadC*)[10]. Instead, biofilm populations accumulated mutations in type 1 fimbriae tip lectin FimH promoting biofilm formation[50], which, in turn, promoted biofilm-associated tolerance. Furthermore, the evolution of antibiotic resistance to amikacin correlated with the early selection of *sbmA* and *fusA* mutations in the biofilm environment. Our study therefore shows that, when biofilms are exposed to clinically relevant intermittent lethal antibiotic treatments[51,52], enhanced biofilm tolerance combined with high mutation rates leads to the rapid selection of clones with increased MIC to amikacin. This suggests that biofilms provide a heterogenous environment where higher mutation rates and intrinsic tolerance promote the survival of otherwise counter-selected resistance mutations, potentially contributing to the antibiotic crisis. Our study may inform the modification of current biofilm infection treatments to reduce the risk of antibiotic resistance emergence and disease recurrence by providing new insights into the evolution of antibiotic resistance in biofilms.

## Results

**Biofilms evolve enhanced survival to intermittent lethal antibiotic treatment faster than planktonic populations.** To investigate the potential emergence of antibiotic resistant mutants in biofilm populations formed by the pathogenic *E. coli* strain LF82, we used a protocol mimicking the repeated treatments of biofilm-associated infections with amikacin, an antibiotic that is recommended for the treatment of Enterobacterales infections[46,53,54]. We exposed planktonic cultures or biofilms formed on medical-grade silicone coupons to the same concentrations of 5xMIC (80 μg/mL) or 80xMIC (1280 μg/mL) amikacin during ten cycles of 24 h intermittent treatments (Fig. 1a, b). Both treatments correspond to concentrations that are lethal and superior to the mutation prevention concentration (MPC: 64 μg/mL). We showed that biofilms treated with such periodic amikacin treatments displayed enhanced survival compared to planktonic populations (supplementary Fig. 1). However, we observed that the percentage of surviving biofilm bacteria showed an initial drop following the first cycle of 24 h treatment, suggesting a relatively low level of biofilm maturity and tolerance. After this first cycle, biofilm cell survival rapidly recovered to reach almost 100% of the population as early as cycle 2 or 3 for the 5xMIC

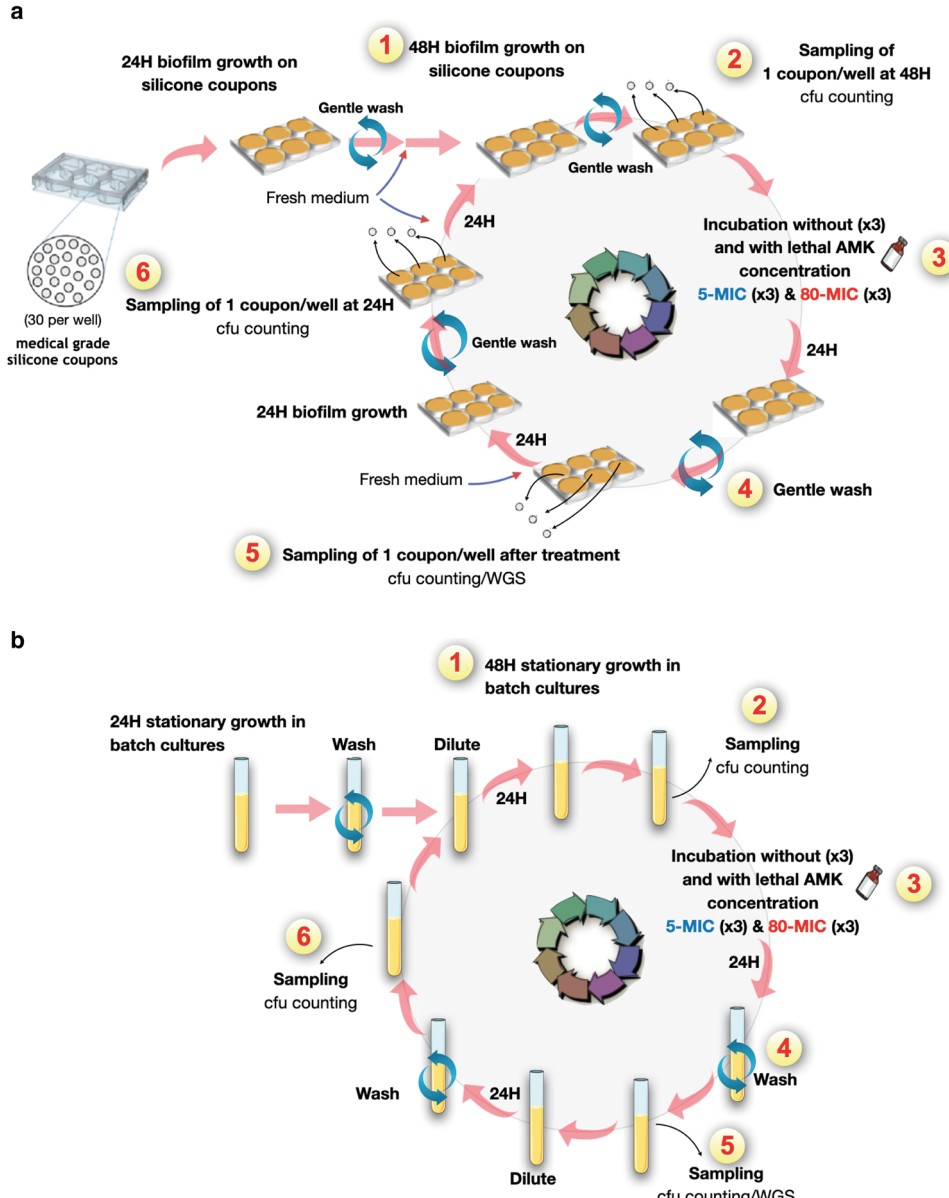

**Fig. 1 Schematics of experimental evolution in biofilm and planktonic populations under intermittent amikacin exposure with lethal concentrations.**
**a** In vitro biofilm-model on silicone coupons. (1) Biofilms formed by *E. coli* LF82 strain are grown for a total of 48 h at 37 °C on silicone coupons in a 6-well plate under static conditions (=cycle 0). (2) One coupon is sampled from each well after removing the exhausted medium and gentle wash of the wells twice with fresh LB medium. (3) Biofilms on silicone coupons are treated with or without amikacin (AMK) at 5- or 80-fold MICs for 24 h at 37 °C. (4) Each well is gently washed twice with fresh LB medium after removing the supernatant containing AMK. (5) One coupon is sampled from each well. Next, the survived biofilm cells are incubated in fresh LB medium for 24 h at 37 °C. (6) One coupon is collected after washing the wells. Then, biofilms are grown for another 24 h at 37 °C (=step 1). Steps 2–6 are defined as a series of cycles, and the experiment is performed for 10 cycles. The last cycle was stopped at step 5. Each coupon is collected in a microtube containing LB medium, and the microtubes are vortexed and sonicated to detach biofilm cells from silicone coupons. The bacterial number of collected sample was confirmed by performing CFU counts. Each collected sample is stored at −80 °C and characterized later. **b** In vitro planktonic growth. (1) After washing and diluting aliquots of the 24-h pre-incubated culture of LF82 strain, the aliquots are incubated in fresh LB medium into a test tube for another 24 h at 37 °C under shaking condition (=cycle 0). (2) Aliquots of the incubated culture are collected for further analyses. (3) The planktonic culture is treated with or without AMK at 5- or 80-fold MICs for 24 h at 37 °C under shaking condition. (4) Aliquots of treated planktonic cells are washed twice with fresh LB medium. (5) Some aliquots of the washed samples are collected. The bacterial number of collected sample was confirmed by performing CFU counts. Next, the remaining aliquots are diluted 100-fold in fresh LB medium and incubated for 24 h at 37 °C under shaking condition. Then, the 24 h incubated culture is washed twice with fresh LB medium. (6) Aliquots of washed samples are collected. Next, the remaining aliquots are diluted in fresh LB medium and incubated for another 24 h at 37 °C under shaking condition (=step 1). Similar to the biofilm experiment, steps 2–6 are defined as a series of cycles, and the planktonic evolution experiment is performed for 10 cycles. The last cycle was stopped at step 5. Each planktonic sample is stored at −80 °C and characterized later, as well as biofilm samples. See also the corresponding Methods section. Some icons of this figure were extracted from BioRender.

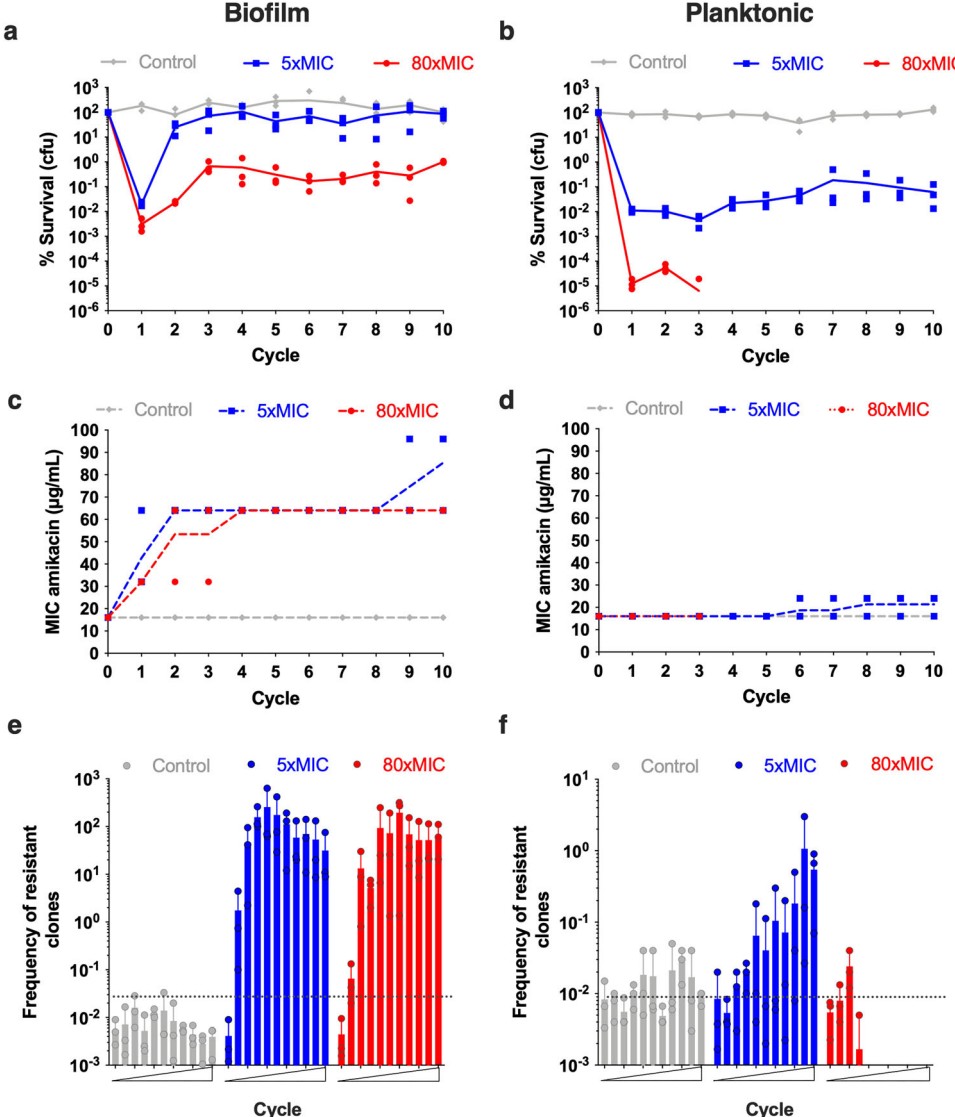

**Fig. 2 Evolution of _E. coli_ survival under lethal antibiotic 24 h intermittent treatment of amikacin. a**, **c**, **e** Biofilms and b, d, f Planktonic. In **a**, **b** are represented the percentage of survival after each cycle of evolution at step 5 for biofilm and planktonic population compared to population before each cycle of treatment at step 2 (see Fig. 1). Percentages of survival were determined using CFU counting, (CFU$_{step5}$/CFU$_{step2}$) × 100. 100% (10$^2$%) survival corresponds to no death after treatment. Three independent evolutions were performed in parallel for each condition (no treatment = control, 5x AMK MIC, 80x AMK MIC). Lines represented correspond to the means of three measurements represented as individual dots ($n = 3$), and are represented on a log scale. In **c**, **d** the MIC to AMK was determined in triplicate ($n = 3$) for each population sampled at the end of each cycle using the agar dilution method. No variation of the MIC was observed between the triplicate. MICs are represented by lines corresponding to the mean of each MIC for the three independent evolutions for each condition ($n = 3$). In **e**, **f** each population sampled at the end of each cycle was plated once on LB plate with or without 1x AMK MIC (or 2xMIC and 4xMIC, see supplementary Fig. 4). Frequency of resistant clones was calculated as the CFU$_{1xMIC}$/CFU$_{LB}$. Values represented correspond to the means and standard errors of the mean of three measurements ($n = 3$), and are represented on a log scale. Representation of the behaviour of individual population is given in supplementary Fig. 2. In a to f, control in grey corresponds to the evolution where no antibiotic was added during step 3.

treatment and around 1% of the population for the 80xMIC treatment (Fig. 2a and supplementary Fig. 2a). By contrast, the planktonic population did not recover after the first cycle of amikacin treatment. At the higher 80xMIC concentration, none of the three parallel replicates showed any survivors after 3 cycles, while only 0.1% of survival was observed after 7–10 cycles with amikacin 5xMIC (Fig. 2b and supplementary Fig. 2b). These results demonstrated that the 80xMIC treatment led to increased killing compared to 5xMIC for both biofilm and planktonic populations, and that biofilms are much more resilient to amikacin than planktonic populations, with an increase of the biofilm population survival during repeated

treatment, which was not observed for treated planktonic populations (supplementary Fig. 3).

**Intermittent antibiotic treatment leads to rapid MIC increase in biofilm but not planktonic populations**. To investigate the genetic events underlying the differences in survival observed between treated biofilm and planktonic populations, we determined the evolution of their MICs at each treatment cycle. We showed that the MICs increased following the population survival profiles, with a net and rapid MIC increase in treated biofilms, but only a moderate and late increase in treated planktonic

populations (Fig. 2c, d and supplementary Fig. 2c and 2d). Consistently, the frequency of resistant clones when populations were plated on 1xMIC, 2xMIC and 4xMIC amikacin increased more rapidly and at lower MIC in treated biofilm populations compared to treated planktonic ones (see Fig. 2e, f, supplementary Fig. 2e and f for 1xMIC/2xMIC and supplementary Fig. 4 for 4xMIC). Moreover, compared to planktonic populations, the frequency of clones from biofilm populations on plates with 2xMIC and 4xMIC was much higher, while no planktonic clones grew on plates with 4xMIC (supplementary Fig. 4). This demonstrated that the evolved clones originating from biofilms displayed a much higher antibiotic resistance level than planktonic ones. It should also be noted that resistance evolved in both biofilms and planktonic populations despite the use of above MPC antibiotic concentrations.

Finally, we randomly isolated clones from cycle 10 of each evolved population, grew them as planktonic cultures and biofilms, and assessed their survival upon treatment with 5xMIC or 80xMIC amikacin. We observed that most of them displayed enhanced survival to 5xMIC or 80xMIC amikacin as compared to their non-evolved parental strain (supplementary Fig. 5). This therefore demonstrated a correlation between the increased survival of biofilm clones and the one of the corresponding evolved populations. Moreover, clones isolated from biofilms displayed higher survival rates when grown as biofilms as compared to when grown as planktonic. This suggested that, beyond the genetic adaptation of these clones, the biofilm environment, per se, plays a critical role in the observed increased survival rates. Despite being propagated with a lower number of generations (supplementary Data 1), biofilms therefore evolve resistance faster and at higher frequency than planktonic populations.

**The rapid evolution of antibiotic resistance in biofilm populations correlates with the early selection of mutations in *sbmA*, *fusA* and *fimH* genes.** To identify the genetic determinants of increased survival and MIC observed upon intermittent antibiotic treatment, we sequenced the end-point biofilm and planktonic populations at cycle 10 exposed or not to amikacin at 5xMIC, as well as biofilm populations exposed to amikacin at 80xMIC (Fig. 3 and supplementary Fig. 6 and supplementary Data 2). The planktonic populations exposed to amikacin at 80xMIC did not survive after 3 cycles and thus could not be sequenced at cycle 10. End point populations displayed up to 9 different mutations with frequency >5%, relative to the ancestral population. We identified mutations in *sbmA* in all treated biofilm populations and 2 out of 3 treated planktonic populations, but not in control ones. SbmA is an inner-membrane peptide transporter, which loss of function was previously associated to increased *E. coli* resistance to aminoglycosides, including amikacin[55–57]. Most populations with *sbmA* mutations displayed several *sbmA* mutated alleles, including stop codon mutations, at various frequencies, indicating potential clonal interference between these different alleles (supplementary Fig. 6 and supplementary Data 2). This was particularly evident in biofilm evolved populations, where 5 out of 6 evolved populations displayed 2 to 5 *sbmA* mutated alleles, regardless of treatment concentration.

In addition, evolved biofilm population 2 exposed to 5xMIC of amikacin displayed a mutation in *fusA*. *fusA* encodes the essential elongation factor G, which mutation was also previously associated to increased resistance to aminoglycosides including amikacin[55–59]. Finally, almost all evolved biofilm populations displayed mutations in the gene coding for the FimH tip-adhesin of type 1 fimbriae at a higher frequency than in planktonic populations[60]. Although these mutations are not directly

associated to antibiotic resistance, they could increase biofilm formation and therefore improved survival against antibiotics (see below in section related to FimH).

The sequencing of populations at intermediate cycles revealed that the emergence of different *sbmA* alleles in biofilm populations was not only more rapid but also very dynamic, with increased diversity of alleles as compared to planktonic populations (Fig. 4 and supplementary Data 3). Moreover, we confirmed the link between presence of *sbmA* mutations and enhanced population resistance: planktonic populations P1 and P2 contained mutations in *sbmA* and displayed an increased MIC towards amikacin, whereas planktonic population P3 had no mutation in *sbmA* and no such increase (Figs. 3 and 4). Population P3 also had the lowest number of colonies growing on amikacin containing plates (supplementary Fig. 2). We confirmed the presence of mutations in *fusA* at intermediate cycles in biofilm populations and showed that, in two populations, *fusA* P610L (population B1) and *fusA* G604V (population B4) mutations reached above the 5% frequency detection threshold at cycle 4, while the *fusA* G604V mutation (population B2) was detected as early as cycle 1 and maintained itself at high frequency in the population until the end of the experiment (Fig. 4 and supplementary Data 3). At the population level, *fusA* and *sbmA* were not mutually exclusive and could coexist in biofilm populations (i.e., in population B1 and B2) but *fusA* mutations were detected before (population B2 and B4) or at the same cycle than *sbmA* mutations. No *fusA* mutation could be detected at frequency >5% in evolved planktonic populations.

Our analyses therefore showed that intermittent treatment with lethal concentrations of amikacin led to enhanced antibiotic resistance reflected by an increase in amikacin MIC and frequency of resistant clones. This demonstrated that the early selection of mutations in *sbmA* and *fusA* are the main drivers of the observed enhanced biofilm survival. By contrast, the weak enhanced resistance of evolved planktonic population emerging in late treatment cycles correlates with rare *sbmA* mutations at a relative lower frequency.

**The biofilm lifestyle and intermittent amikacin treatment led to convergent selection of mutational diversity in *sbmA* and *fusA* genes in evolved clones.** The diversity of mutations at the clone level was further examined by sequencing the end-point clones (see supplementary Fig. 5) as well as additional randomly chosen clones originating mostly from cycle 10 (Fig. 5 and supplementary Data 4). We first observed that most evolved clones originating from both biofilm and planktonic populations exhibited increased MIC compared to the ancestor strain (16 µg/mL) (Fig. 5 and supplementary Data 4). This was especially the case for clones carrying single mutations in *fusA* (clones 853, 863 and 891 with single *fusA* mutations G604V, G676C and P610L, respectively) or *sbmA* (clones 236 and 501 with single *sbmA* mutations Q192L and coding (17/1221 nt) (T)6 → 7, respectively) and otherwise isogenic compared to the ancestral strain. This allows for the direct correlation between enhanced MICs and specific mutations. In biofilm evolved populations, clones presenting a moderate MIC increase of up to 24 µg/mL of amikacin carried a loss of function mutation in the *sbmA* gene only, or mutations not previously linked to antibiotic resistance. By contrast, clones exhibiting the highest MIC (up to 128 µg/mL) carried *fusA* mutations, all located in domains IV and V of the elongation factor G protein (supplementary Fig. 7 and supplementary note). Indeed, the presence of single (G604V, P610L and G676C) or double (A597V P610L) *fusA* mutations led to a MIC of 48 µg/mL (Fig. 5). The highest MIC (128 µg/mL) was reached in clones with a mutation in *fusA* and additional mutations in: (i) *sbmA* and *fre* NAD(P)H flavin reductase encoding gene, which was previously shown to enhance

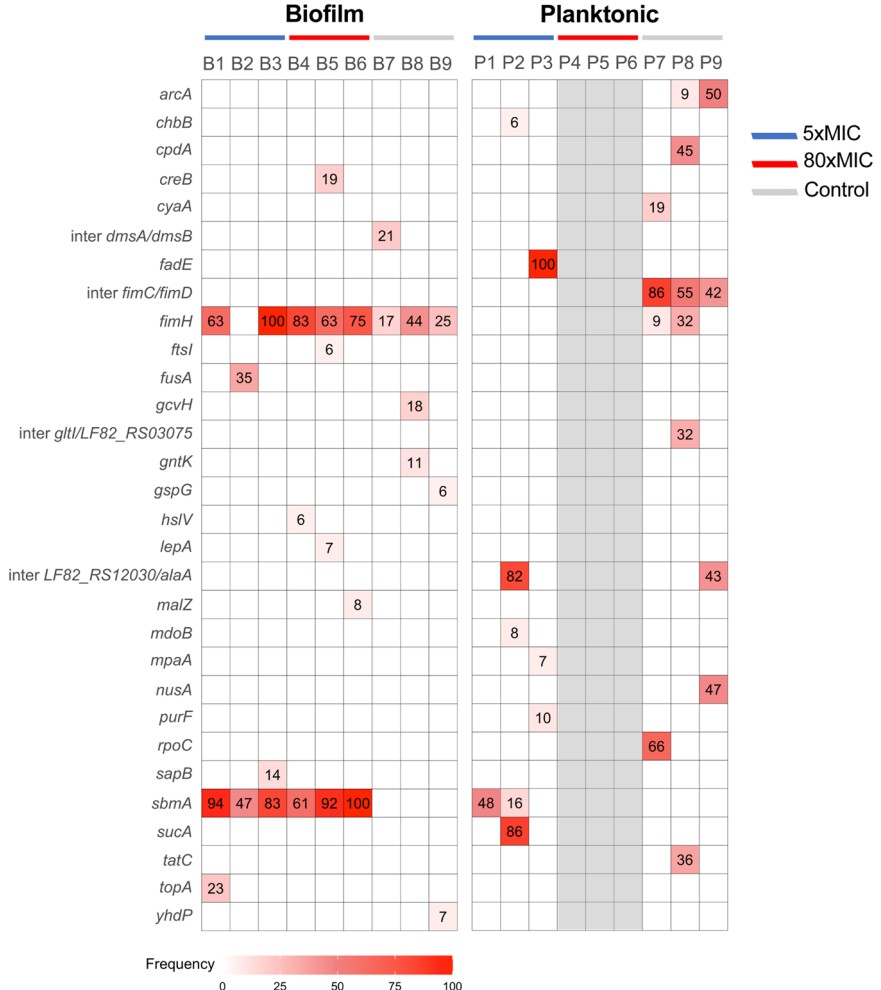

**Fig. 3 End-point population sequencing reveals a lifestyle associated pattern of mutations after evolution under intermittent antibiotic treatment.**
Mutations identified by whole-population genome sequencing of control and AMK treated (5xMIC and 80xMIC) biofilm and planktonic populations of *E. coli*. The three populations corresponding to the three evolved lineages per lifestyle and treatment were sequenced after 10 cycles of treatment. The average depth of sequencing was x150. Mutations at higher frequency than 5% are detected by Breseq analysis. Red shading indicates the total frequency of all mutations in each gene within a population at cycle 10 of the experimental evolution and the number in each box corresponds to the exact frequency detected. Population 4, 5, 6 from the planktonic lifestyle did not survive after 3 cycles and thus could not be sequenced at cycle 10 (they are shaded in grey). We therefore sequenced the population corresponding to the last cycle before their respective extinction, but no mutation was detected at frequency of 5% or higher, with the exception of one fixed mutation in population P5 (see supplementary Data 2 and 3).

resistance to amikacin[55–59] or (ii) *yfg*Z, a gene encoding a protein involved in oxidative stress repair and Fe-S cluster synthesis[61,62]. The *yfgZ* gene is also potentially involved in aminoglycoside resistance[63] and tRNA modification, since a *ygfZ* null mutant has lower levels of several modified nucleotides in tRNA[61], thus possibly linking increased mistranslation with enhanced resistance to some antibiotics[64,65]. Interestingly, whole genome and Sanger sequencing of the *sbmA* and *fusA* genes in clones from intermediate cycles indicated that the presence of *fusA* mutations could be detected at early cycles of biofilm evolution, suggesting that they were probably already present at low frequencies in the original populations (supplementary Data 4 and 5).

The analysis of clones isolated from evolved planktonic populations revealed the presence of *sbmA* mutations, similar to evolved biofilm clones but with less diversity (Fig. 5 and supplementary Data 4). However, planktonic evolved clone MICs also correlated with a higher variety of other mutations compared to the ones found in clones from evolved biofilm populations (see supplementary note). These other mutations were nevertheless not detected at frequency >5% in any of the sequenced populations

(supplementary Data 3). Comparison of the MICs of the different clones isolated from biofilm vs planktonic evolution indicated that the MICs of biofilm clones was higher than planktonic clones (supplementary Data 4, 5 and 6 and supplementary Fig. 8). This supported our previous observation that evolution in biofilm conditions generated clones with higher resistance than evolution in planktonic conditions.

These results showed that, at the clone level, the biofilm lifestyle, combined with intermittent amikacin treatment, led to the convergent selection of diverse mutations in *sbmA* and mutations in *fusA*. However, except for mutations in *sbmA*, the pattern of selected mutations differed between biofilm and planktonic evolved populations. Planktonic evolved populations were associated with mutations in a more diverse set of genes with lower MICs than those observed in evolved biofilm populations.

**The high biofilm mutation rate favours the emergence of amikacin resistance in biofilms despite the fitness cost of *fusA* mutations.** The absence of detectable *fusA* mutations seemed

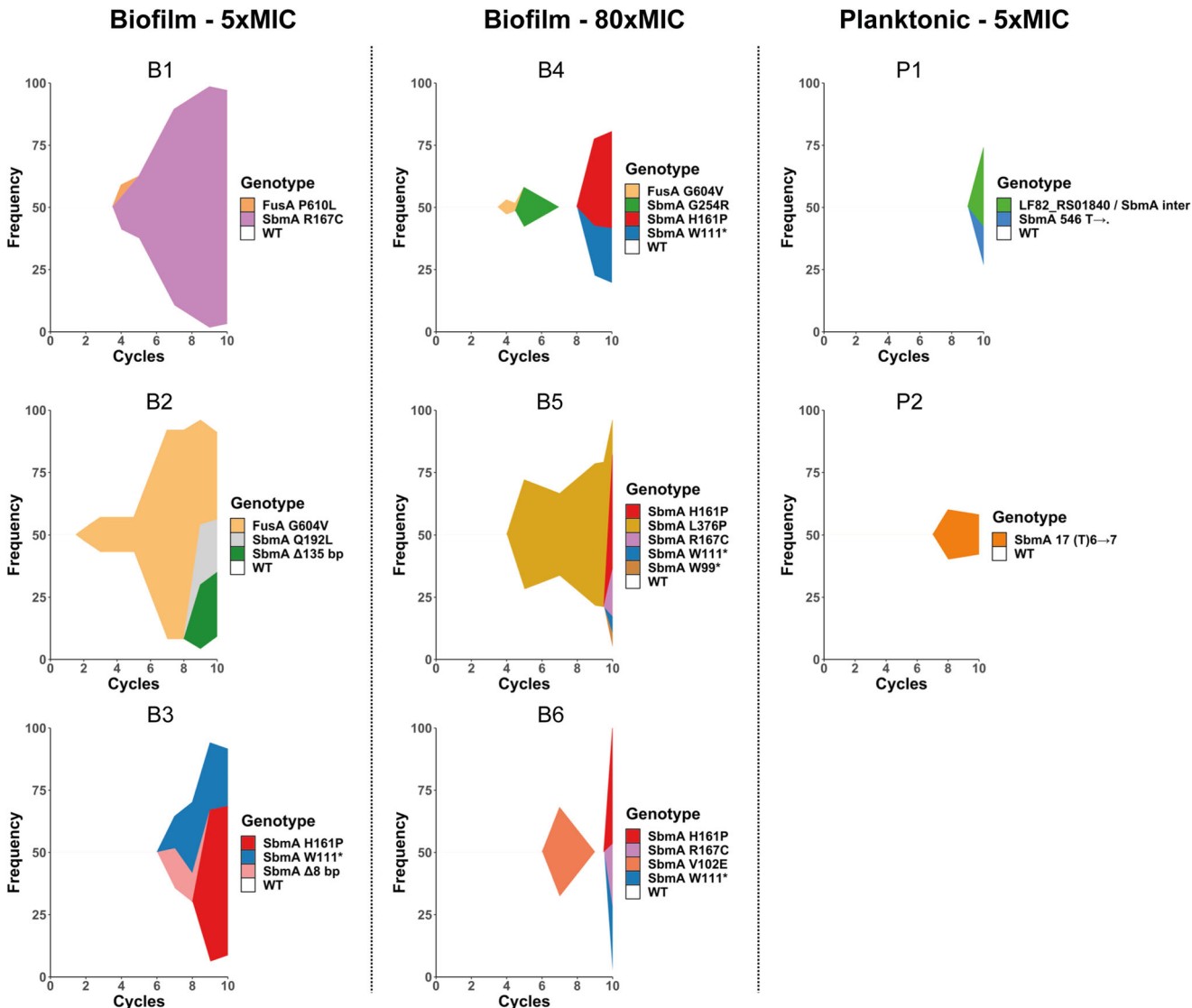

**Fig. 4 Muller plots showing the dynamics of *sbmA* and *fusA* allele frequencies during the evolution of biofilm and planktonic populations subjected to intermittent amikacin treatment.** An independent muller plot showing the dynamic of *sbmA* and *fusA* alleles frequencies over cycles of evolution is shown for each population. Every mutation is displayed with a unique colour through all plots. Due to the limit of detection, only mutations with frequency above 5% frequency are shown. Information regarding *sbmA* and *fusA* have been extracted from whole-population sequencing presented in supplementary Data 3.

to be characteristic of evolved planktonic populations. To test whether *fusA* mutations could be present at very low frequency in planktonic populations, glycerol stocks of evolved planktonic populations were grown overnight and concentrated 10 times before being plated on 2x amikacin MIC plates. Among the clones growing on these plates, we could indeed detect a high diversity of mutations in *fusA*, even in the early evolution cycles (supplementary Fig. 7 and supplementary Data 6). This confirms the presence of *fusA* mutants at very low frequencies at early cycles, which were not selected and maintained in further selection cycles. To investigate why *fusA* mutations were selected in biofilm but not planktonic populations, we determined *sbmA* and *fusA* mutant growth capacity as well as their fitness cost in planktonic conditions. Only P610L *fusA* mutants displayed a relatively lower doubling time (supplementary Fig. 9). However, all three identified *fusA* mutants exhibited a higher fitness cost than the *sbmA* mutants in absence of antibiotics, and this cost was still present when the *fusA* mutants were regrown after amikacin treatment (Fig. 6). This suggested that *fusA* mutants could have been counter-selected during intermittent planktonic growth between

treatments, while being protected from such counter-selection and maintained in biofilm populations.

Alternatively, or additionally, *fusA* mutations could have accumulated at higher frequency in the stressful and mutation-prone biofilm environment[34,66]. Consistently, we showed that the mutation rate in the biofilms formed in our experimental evolution set-up was higher than in planktonic conditions (supplementary Fig. 10), therefore providing increased opportunity for genetic diversification and the rapid apparition and selection of resistance mutations.

**Mutations enhancing biofilm formation also contribute to enhanced biofilm survival.** We recently showed in a previous study that mutations enhancing adhesion to abiotic surfaces are rapidly selected in the type 1 fimbriae tip lectin FimH during biofilm formation, including in frame deletions[50]. Considering the increased intrinsic tolerance of biofilm to antimicrobials, we hypothesized that the high frequency of mutations identified in *fimH* in our evolved biofilm populations —all located in the

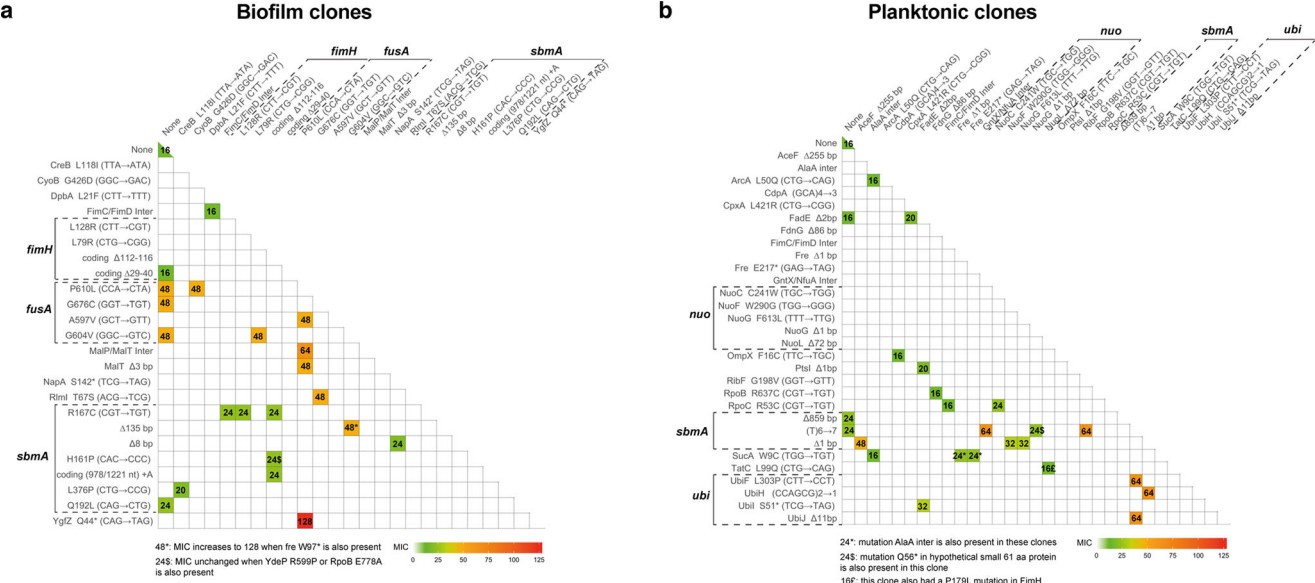

**Fig. 5 Association of non-synonymous mutations in sequenced evolved clones and their corresponding MICs to amikacin.** Non-synonymous mutations identified by whole genome sequencing in clones of *E. coli* coming from AMK intermittently treated biofilm (**a**) and planktonic (**b**) populations. MICs are expressed in µg/mL. MIC of the ancestor strain is 16 µg/mL and is shown in the first top left cell. In **a** and **b** the first column actually corresponds to clones that had a unique mutation in the indicated genes. The presented information is extracted from clone sequencing data presented in supplementary Data 4.

FimH lectin domain (supplementary Fig. 11)— could also contribute to increase biofilm tolerance and survival (Fig. 3). To test this, we compared two of the obtained *fimH* mutants (in frame deletion Δ29-40 and L79R) with their otherwise isogenic wild type alleles and showed that these *fimH* mutations indeed promoted biofilm formation (Fig. 7a, b) as well as survival when exposed to 80xMIC of amikacin (Fig. 7c). Therefore, the differential accumulation of *fimH* mutations in evolved biofilm populations also likely contributed to enhanced tolerance to otherwise lethal antibiotic concentrations.

## Discussion

The high but transient tolerance of biofilms to antibiotics and host immune defenses is a well-recognized cause of bacterial chronic infections[67–69]. Although the risk of emergence and evolution of genetically inheritable antibiotic resistance in bacterial biofilms is likely underestimated, such events have been described during the long-term or repeated antibiotic treatments of chronic biofilm-related infections, suggesting that these treatments could promote the emergence of antibiotic resistance[70]. The most documented cases correspond to longitudinal analyses of *P. aeruginosa*, *S. aureus*, and *Burkholderia* sp. clones isolated from cystic fibrosis patients subjected to lifelong antibiotic treatments[71–77]. The emergence of vancomycin, rifampicin or daptomycin resistance has also been reported during *S. aureus* endocarditis or catheter-related bacteremia treatments[78–81]. Moreover, within-patient antibiotic resistance evolution has been described for various bacterial pathogens, including *E. coli* or *Mycobacterium tuberculosis*[82–84], where infection was suspected to originate from biofilms[85–89]. In this study, we used adaptive experimental evolution to show that biofilm populations exposed to periodic treatment with lethal doses of the aminoglycoside antibiotic amikacin, spaced with periods allowing biofilm regrowth, developed antibiotic resistance much faster and at higher levels than planktonic populations.

Previous in vitro studies of the evolution of antibiotic resistance within biofilms used either constant sub-MIC[40–43], stepwise increased concentrations starting from sub-MIC[39,42,44] or constant lethal concentration[36]. Our experimental settings, instead,

used periodic treatment with lethal concentrations above the MPC. In both planktonic and biofilm conditions, these periodic regimens generated, at the population level, convergent mutations in genes related to antibiotic resistance, including *fusA* and *sbmA*. These results are in agreement with the notion that lethal antibiotic concentrations lead to the stringent selection of a few selected mutations, therefore increasing the MIC in a single genetic event[90]. However, the characteristic increased diversity of mutations observed in previous experimental evolution experiments was, detected at the allelic level in our case, with up to five mutated alleles in the gene encoding the peptide transporter SbmA within the same end-point biofilm population. It should be noted that the evolution of resistance in our experiments, despite the use of antibiotic concentrations above the MPC, can be explained by the large populations present in our settings. This may have led to the absence of full eradication (notably in biofilms) and the capacity of bacteria to resume growth and mutate during the recovery phase between treatment cycles.

Another difference observed between evolved biofilm and planktonic populations was the detection of *fusA* mutations coding for the elongation factor G, an essential protein that facilitates the translocation of the ribosome along the mRNA molecule. *fusA* mutations were found at frequency >5% in 3 out of six biofilm populations exposed to amikacin. In one of them, a *fusA* mutation was detected early and maintained throughout all evolution cycles. At the clonal level, *fusA* mutations were also detected in planktonic populations, albeit only after concentrating the samples. This suggests that these mutations were rarely present but were not selected in planktonic populations, while being selected and maintained in biofilm populations. All *fusA* mutations were exclusively located in domains IV and V of the FusA protein. This bias towards these two domains was also observed in other evolution experiments of *E. coli* in the presence of aminoglycosides[55,56,58,59] (see supplementary note and supplementary Fig. 13). Surprisingly, convergent *fusA* mutations detected in both planktonic and biofilm populations of *A. baumannii* and *P. aeruginosa* treated by stepwise increasing concentrations of tobramycin[44] did not display any apparent domain bias, suggesting that the functioning of this FusA factor might

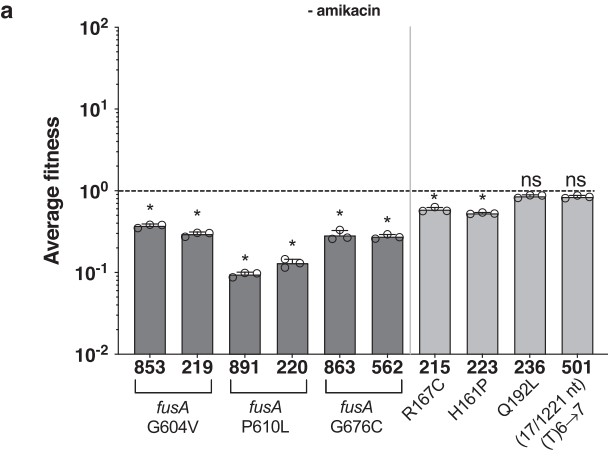

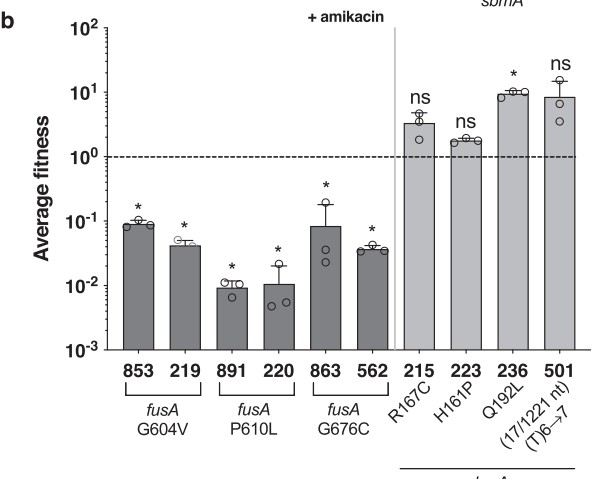

**Fig. 6 Relative fitness costs of evolved clones containing *fusA* and *sbmA* mutations.** Each evolved clone tagged with mars was competed with wildtype tagged with GFP at a 1/1 ratio in the absence (**a**) and presence (**b**) of 5x AMK MIC. In order to match the evolution protocol, in (**a**) wt/mutants LB overnight cultures were mixed, diluted to OD 0.02 and grown in LB for 24 h before flow cytometer analysis. In **b** wt/mutants LB overnight cultures were mixed, diluted to OD 2, treated for 24 h by 5x AMK MIC, washed, diluted 1/100 and regrow in LB for 24 h before flow cytometer analysis. The fitness index of the comparison of wildtype tagged with mars to wildtype tagged with GFP is taken as 1 (dotted line). Detailed information on the different analysed clones can be found in supplementary Data 4. It should be noted that clones 853, 863 and 891 differ, respectively, from the ancestral strain by a single G604V, G676C and P610L *fusA* mutation, and clones 236 and 501 differ, respectively, from the ancestral strain by a single Q192L and (17/1221 nt) (T)6 → 7 *sbmA* mutation, and are otherwise isogenic. The data corresponded to n = 3 biological replicates for each clone and are represented as the mean ± SD. Statistics correspond to unpaired two-tailed *t*-test with Welch's correction comparing each condition to wildtype/wildtype fitness. ns, no significance; *$p < 0.05$; **$p < 0.01$; ***$p < 0.001$.

differ between species. Interestingly, the alignment of all the *E. coli fusA* sequences in the nrprot database of NCBI (2143 sequences) identified 11 *fusA* sequences with the A678V mutation and 6 sequences with the P610S mutation (supplementary Data 7), therefore suggesting that the selection of *E. coli fusA* mutations could bear relevance beyond our in vitro experiments.

One major question raised by our study is why biofilms accumulated resistance mutations so rapidly. This could be due to the well-described biofilm-associated oxidative and chemical stress response leading to enhanced mutation rate[30,31,91], also observed in our experimental settings. Alternatively, but not exclusively, the high frequency of bacteria possessing different mutations that increased the MIC to amikacin within biofilm populations could also explain their higher capacity to survive lethal amikacin concentration. This reflects the propensity of the biofilm environment to favour clonal interference compared to planktonic conditions[34]. Biofilm intrinsic tolerance to lethal antibiotic concentration as high as 80xMIC also increases the size of surviving bacterial population within biofilms at each cycle of treatment, while planktonic populations were eliminated after three cycles of such treatment. This observed intrinsic tolerance of biofilms could also be partly explained by the possible reduced penetration of amikacin into biofilms as has been shown for *P. aeruginosa* biofilms[92] and as we showed for *E. coli* LF82 colony biofilms (supplementary Fig. 12).

Enhanced survival to antibiotics is also likely to play a role in the biofilm capacity to shelter mutants that were selected early for their antibiotic resistance. Interestingly, such a relationship between tolerance and resistance, with tolerance promoting the emergence of resistance, has also been demonstrated in planktonic populations treated intermittently with antibiotics[11,12,14]. However, in our case, tolerance was not provided through selection of de novo induced genetic mutations previously reported to increase tolerance in periodically aminoglycoside treated *E. coli* planktonic populations (i.e., in *nuoN*, *oppB* or *gadC*)[10], but by the intrinsic antibiotic tolerance of biofilms. If these tolerance mutations did emerge, the stringency and length of the antibiotic treatment used in our study (24 h) could have prevented their selection, by contrast with previous studies using shorter treatment (maximum of 8 h)[13].

The outcome of our experimental evolution experiments is that biofilm evolved clones displayed higher resistance than evolved planktonic ones. This is in stark contrast with studies exploring biofilm evolution under constant antibiotic pressure that show the emergence of mutants with lower resistance level compared to evolved planktonic populations[41,42]. The use of lethal concentrations of amikacin in our intermittent treatment logically favoured the selection for high resistance mutations in both lifestyles[90], including *fusA* mutations, conferring resistance to up to 3 times the amikacin MIC. However, these mutations were rapidly counter-selected in planktonic populations, while being maintained in biofilms. We showed that *fusA* mutations are associated with a strong fitness cost in planktonic conditions, suggesting that they could be rapidly eliminated in planktonic conditions, notably during the regrowth period without antibiotic. The capacity of structured biofilm environments to maintain high-cost mutations was observed in several biofilm evolution studies[36,41,42,45], in which rare beneficial mutations could be protected from competitions in biofilm micro niches, thereby increasing their odds to be selected and propagated. In support of this hypothesis, it was recently shown that the SOS response induced in *E. coli* biofilm micro niches leads to increased rate of mutagenesis[93] and that beneficial resistance mutations had reduced chances to be lost by genetic drift in structured environments[94]. Alternatively, due to diffusion limitation, the concentration of antibiotics could drop more slowly in biofilms than in planktonic conditions and traces of antibiotics present in biofilms between each cycle of antibiotic treatment could reduce the fitness cost of resistance mutations by providing a compensatory growth advantage.

Whereas we did not detect mutations previously associated with increased antibiotic tolerance, we identified, in addition to bona fide aminoglycoside resistance mutations in *fusA* and *sbmA*, several mutations in *fimH*, the gene encoding the tip-lectin of the type 1 fimbriae. We recently showed that, in dynamic

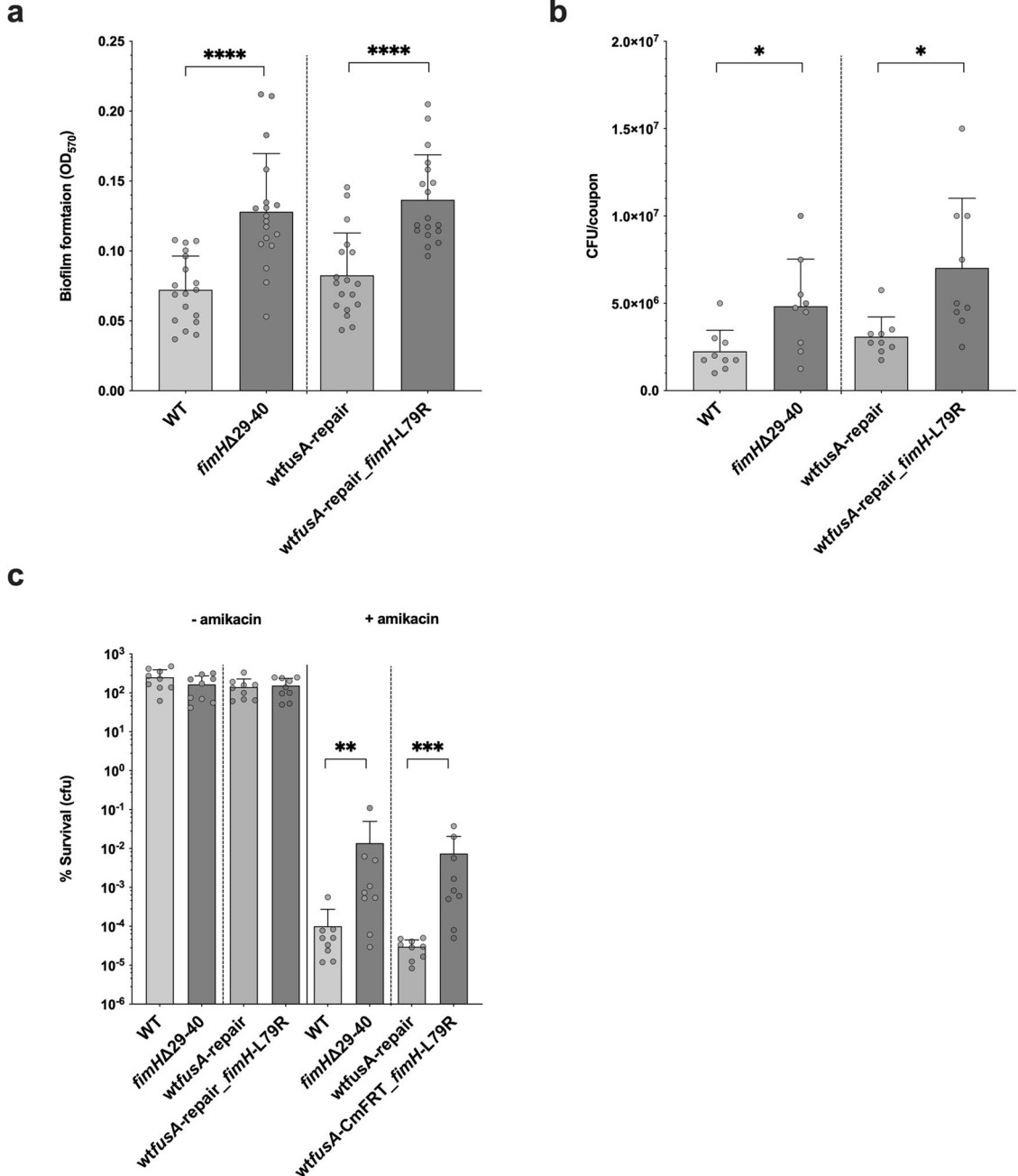

**Fig. 7 *fimH* mutants display higher capacity to form biofilms and enhanced tolerance towards a lethal concentration of amikacin. a, b** Biofilm amounts (**a**) and colony forming units (**b**) of *fimH* mutants formed on silicone coupons for 24 h. **c** The survival rate of biofilm cells of *fimH* mutants on silicone coupons after the treatment with 80xMIC of AMK. It was calculated by using the formula 100x(CFU$_{AFTER}$/CFU$_{BEFORE}$), where CFU$_{BEFORE}$ corresponds to CFU/coupon before treatment and CFU$_{AFTER}$ to CFU/coupon after treatment. The data corresponded to $n = 18$ biological replicates for a, and $n = 9$ for **b** and **c**. Data are represented as the mean ± SD. Statistics correspond to unpaired two tailed *t*-test with Welch's correction comparing each *fimH* mutant to its corresponding wt isogenic strain. *$p < 0.05$; **$p < 0.01$; ***$p < 0.001$, ****$p < 0.0001$. In **c**, statistics were performed on log-transformed % survival. *fimH* Δ29-40 in frame deletion is present in clone 239 that differs from the ancestral strain by this single mutation. We showed previously that the exact same *fimH* Δ29-40 deletion enhanced biofilm formation in *E. coli* K12[50] (see supplementary Data 4). The clone with the mutation *fimH* L79R was created after de-construction of the *fusA* G604V mutation from the clone 219 where we replaced the *fusA* G604V allele by a wt version of *fusA* (wt*fusA*-repair) (see Methods). This clone also codes for a synonymous mutation I255I in the gene *LF82_RS22000*.

continuous flow conditions, *fimH* is under strong positive selection for mutation promoting non-specific *E. coli* adhesion to surfaces and biofilm formation[50]. Mutation in FimH tip-lectin domain identified in the present study not only enhanced *E. coli* biofilm formation, but also its survival to lethal high concentration of amikacin. Such enhanced biofilm capacity has also been observed in (i) *A. baumannii* bead biofilms constantly exposed by

sub-MIC or step-wise increased concentration of ciprofloxacin[42], as well as on (ii) biofilms formed in silicone tubing treated for 3 days with tetracycline[40], and with (iii) *E. faecalis* biofilms exposed to increasing concentrations of daptomycin in turbidostats[39]. While our experimental set-up did not use continuous flow, the biofilms formed on medical-grade silicone coupons were periodically exposed to washes and flux, probably

promoting the selection of biofilm promoting mutations. The selection of such biofilm-promoting mutations could therefore be seen as contributing to a positive feedback loop, enhancing biofilm formation and *de facto* enhancing tolerance and survival of biofilm bacteria, which would increase the odds for de novo genetic resistance mutations to occur.

Altogether, our study demonstrates that biofilms exposed to lethal intermittent treatment rapidly evolved towards enhanced survival to antibiotics through multiple mechanisms involving intrinsic and promoted tolerance and/or enhanced mutation rates that may favor the rapid selection and maintenance of high cost antibiotic resistance mutations. Whereas this phenomenon might be context dependent, our work contributes to the demonstration that antibiotic tolerant biofilm environments generally promote the worrisome evolution of antibiotic resistance[36,39–44,95]. Encouragingly, however, a recent study showed that using short breaks (no more than 4 h) between the oxacillin antibiotic treatments of *S. aureus* catheter-associated infection increased the efficacy of the treatment, while preventing the emergence of antibiotic resistance[48]. This therefore suggests that improving our current understanding of the evolution of biofilms under antibiotic pressure in various regimens and models could lead to new and clinically relevant therapeutic options to fight biofilm-related infections.

## Methods

**Bacterial strains and growth conditions**. Bacterial strains used in this study are listed in supplementary Table 1. The adherent/invasive AIEC strain *E. coli* LF82 was isolated from a chronic ilea lesion from Crohn's disease patients[96]. Overnight cultures were grown at 37 °C using Miller's LB medium (Thermo Scientific, Rochester, NY, USA) under shaking at 180 *rpm* unless otherwise specified. The following antibiotics were used for strain construction: kanamycin (50 µg/mL) and chloramphenicol (25 µg/mL). All compounds were purchased from Sigma-Aldrich (St Louis, MO, USA).

### Strain construction

*Generation of mutants in E. coli LF82.* We generated the LF82 mutants used in this study by λ-red linear recombination[97]. When required, antibiotic resistance markers flanked by two FRT sites were removed using the Flp recombinase[98]. We verified the integrity of all cloned fragments, mutations, and plasmids by PCR with specific primers and whole-genome sequencing.

*Construction of LF82_mars and LF82_GFP.* Genes coding for the Red (mars) or green fluorescent protein (gfp) were inserted on the chromosome of LF82 replacing the LF82 *ampC* gene (supplementary Table 1). In brief, plasmid pZE1RGFP-KmFRT was constructed by amplification of KmFRT region from plasmid pKD4 and cloning between SalI-HindIII sites of plasmid pZE1RGFP. Plasmid pZE1R-mars-KmFRT was constructed by cloning the pZS*2R-mars plasmid's KpnI-BamHI digested region containing the mars open reading frame between the KpnI-BamHI sites of pZE1RGFP-KmFRT plasmid. Then the DNA fragments of mars-KmFRT and gfp-KmFRTwere amplified from, respectively, pZE1R-mars-KmFRT and pZE1RGFP-KmFRT using long recombining primers (supplementary Table 2). The resulting PCR products, which have ~50 bp-long regions of homology upstream of fluorescent genes and downstream of KmFRT, were introduced at the *ampC* gene by λ-red recombination into strain LF82 using pKOBEG plasmid[97], respectively. Then, ΔampC::mars-KmFRT and ΔampC::GFP-KmFRT mutations were introduced into strain LF82 by P1vir phage transduction, in which the Flp recombinase was used to remove the kanamycin marker using pCP20 plasmid[98].

*Deconstruction of the fusA G604V allele in clone 219.* The CmFRT cassette (*cat* gene) was amplified from strain MG1655ΔyfcV-P::CmFRT[99] using long recombining primers (supplementary Table 1, 2). The resulting PCR product, which has ~50 bp-long regions of homology upstream and downstream of CmFRT start and stop codons, was introduced just after the *tufA* gene that follows the *fusA* gene by λ-red recombination into strain LF82_*mars* using pKOBEGA plasmid[97]. This generates the strain LF82_mars, wt*fusA*-CmFRT. A DNA fragment encompassing wt*fusA*-CmFRT was amplified from strain LF82_*mars*, wt*fusA*-CmFRT using the primers, Up.tufA-CmFRT-5 and Down.tufA-CmFRT-3 (supplementary Table 2). The *fusA* G604V allele of the clone 219 corresponding to LF82_*mars*, *fusA_G604V*, *fimH_L79R*, LF82_RS22000_I255I was deconstructed back to wt by it by the wt*fusA*-CmFRT region using λ-red recombination with the pKOBEGA plasmid.

**MIC and MBC determination**. We determined the minimum inhibitory concentration (MIC) and minimum eradication concentration (MBC) values of amikacin (AMK) (Sigma-Aldrich) for the LF82 strain in triplicate using the micro-broth dilution method[100] with some minor modifications. AMK was diluted at various concentrations, 4, 8, 16, 24, 32, 40, 48, 64, 96, 128 mg/L in sterile 50 µL LB broth in 96 well plate. 50 µL of $10^6$ cell/mL of LF82 was inoculated into each well containing 50 µL of the various concentrations of AMK in LB broth (final concentration 2, 4, 8, 12, 16, 20, 24, 32, 48, 64 mg/L). The plates were incubated at 37 °C for 18 to 24 h and thereafter observed for growth or turbidity (determination of MIC). Subsequently, each well was serial 1 to $10^7$-fold diluted using PBS. 10 µL of dilution were spotted on LB agar plates and incubated at 37 °C for 24 h. The MBC was determined as the lowest concentration killing 99.9% of the bacterial population. The MIC of AMK for LF82 in LB was measured as 16 µg/mL while its MBC was measured at 20 µg/mL. The MIC of AMK for LF82 using Mueller Hinton medium was determined as 4 µg/mL, while K12 strains such as MG1655 had a MIC of 2 µg/mL. LF82 is therefore not resistant to AMK based on EUCAST classification (MIC breakpoint of 8 µg/mL in MH).

We determined the MIC values of AMK or rifampicin (RIF) (Sigma-Aldrich, St Louis, MO, USA) for evolved population and clones using the agar dilution method[100] with some minor modifications. Briefly, evolved clones or populations were inoculated into test tubes containing LB medium and incubated overnight at 37 °C. After diluting the overnight culture to ~$1.5 \times 10^8$ CFU/mL into fresh LB medium, 0.5 µL of the diluted culture was inoculated to three LB agar plates containing AMK or RIF at different concentrations using a Microplanter inoculator ($n = 3$) (Sakuma Factory, Tokyo, Japan). The concentration of AMK was adjusted at a finer level than 2-fold increment (specifically, 4, 8, 12, 16, 20, 24, 32, 48, 64, 128, 256 µg/mL) to increase the sensitivity for finding the susceptibility changing. The concentration of RIF was set conducted in 2-fold increments according to the normal agar dilution method.

**Amikacin concentration dependent killing of *E. coli* LF82 in biofilm and planktonic conditions (measurement of MBIC and MBEC)**. LF82 biofilm and planktonic cells grown in LB for 48 h on silicone disks or in test tubes were treated with or without AMK at 1- to 80-fold MICs (1, 2, 3, 4, 5, 10, 20, 50, 80 and 100-fold MICs) for 24 h, respectively. The survival rate of bacteria under biofilm or planktonic conditions was calculated as the ratio of the number of survived cells after the treatment per the total number of bacterial cells before the treatment by performing CFU counts using bacterial suspensions. Minimum Biofilm Inhibition Concentration (MBIC) and Minimum Biofilm Eradication Concentration (MBEC) were defined as the antibiotic concentration that kill 90 and 99.9 % of biofilm cells, respectively, and corresponded to 48 µg/mL (corresponding to 3xMIC) and 160 µg/mL (corresponding to 10xMIC).

*Biofilm cells on silicone coupons.* LF82 biofilms were grown on 12 silicone coupons sheeted in a well of a 6-well plate (Techno Plastic Products AG) and treated with AMK. Briefly, silicone sheets (Silicone elastomer membrane 7-4107, Dow Corning Corporation, Midland, MI, USA) were cut out into round pieces at a 5 mm diameter using a biopsy punch (Kai Medical, Seki City, Japan). Then, cut coupons were placed on a well in 6-well plate (12 coupons per well), and the plate was sterilized with ethylene oxide. LF82 overnight culture was diluted to an optical density at 600 nm ($OD_{600}$) of 0.05 in LB medium, and 5 mL of the diluted culture was poured into the well. Biofilms were grown at 37 °C for 24 h under static condition. The exhausted medium containing floating cells was then removed from the well using an aspirator (VACUSIP; Integra Biosciences, Hudson, NH, USA), and the well was gently washed once with 5 mL of phosphate buffered saline (PBS, Lonza, Rockland, ME, USA) to remove un-adherent floating bacteria (hereafter, called washing step). Biofilms were refilled with 5 mL of fresh LB medium for an additional 24 h (in total of 48 h). After incubation and washing step, six coupons were aseptically collected using sterilized forceps and each dipped in 500 µL of LB medium in a microtube (= before treatment coupons). The remaining biofilm cells in the well were treated with AMK dissolved in 5 mL of fresh LB medium at 37 °C for 24 h. After removing the medium supernatant, the well was rewashed, and six coupons were collected each in 500 µL of LB medium (= after treatment coupons). Each microtube containing one coupon (before treatment coupons and after treatment coupons) was vigorously vortexed for 1 min and sonicated for 10 min using an ultrasonic bath (Branson 5800, Branson Ultrasonics, CT, USA) to detach biofilm cells from coupons and prepare bacterial suspension. Then, bacterial pellets were harvested by centrifuging at $15,000 \times g$ for 3 min at 25 °C, washed once with 200 µL of PBS, and suspended in 200 µL of LB medium to prepare bacterial suspension. The bacterial number of the suspension was confirmed by performing CFU counts.

*Planktonic cells.* LF82 was incubated in a test tube containing 3 mL of fresh LB medium at 37 °C for 24 h under shaking condition. The culture was adjusted to $OD_{600}$ of 0.01 in LB medium into test tubes and incubated for an extra 24 h (in total of 48 h). After incubation, LF82 culture was treated with AMK at 37 °C for 24 h under shaking condition by directly adding several concentrations of AMK solution in the test tube. Before and after the AMK treatment, 200 µL of the culture was collected in a microtube. Then, bacterial pellets were harvested by centrifuging at $15,000 \times g$ for 3 min at 25 °C, washed once with 200 µL of PBS, and suspended in

200 μL of LB medium to prepare bacterial suspension. The bacterial number of the suspension was confirmed by performing CFU counts. The test was repeated six times for each concentration.

**Mutation Prevention Concentration determination**. We determined the MPC value that prevents the growth of at least $1.0 \times 10^{10}$ bacteria of LF82, according to previous literature[101]. Briefly, 100 mL of LF82 overnight culture was centrifuged at $4500 \times g$ for 10 min at 25 °C. After removing the supernatant, bacterial pellets were suspended in 10 mL of fresh LB and concentrated to ~$2.0 \times 10^{10}$ bacteria/mL. The bacterial number of the plated suspension was confirmed by performing CFU counts on an LB agar plate after 24-h incubation. Subsequently, 100 μL of the bacterial suspension were plated on LB agar plates with or without AMK at 1- to 6-fold MIC and incubated at 37 °C for 72 h. Ten replications were made for each concentration of AMK to evaluate the growth for a total of ~$2.0 \times 10^{10}$ bacteria. The MPC corresponds to the AMK concentration of the plate where no growth was observed.

**Experimental evolution in biofilm and planktonic populations under intermittent amikacin exposure with lethal concentrations**. Evolution experiments, in which biofilm and planktonic populations were intermittently treated with or without AMK at 5- or 80-fold MICs, were performed for 10 cycles. In both conditions, one cycle consists of a 24 h AMK treatment and two 24 h regrowth steps in absence of AMK, with washing steps between 24 h treatment/incubation steps (Steps 2–6 in Fig. 1 were defined as a series of one cycle). Evolved populations and end-point clones obtained from experiments were stocked as glycerol stock at −80 °C and characterized later by MIC testing, the frequency of AMK resistant mutants, survival rate against AMK treatment, and whole-genome sequencing, if necessary.

*Biofilm condition*. First, LF82_GFP and LF82_mars from glycerol stocks were grown overnight in LB medium under shaking condition and were adjusted to OD$_{600}$ of 0.05 in LB medium. After mixing the adjusted cultures of LF82_GFP and LF82_mars at a 1:1 ratio, 5 mL of the mixture was poured into a total of 9 wells (3 wells of a 6-well plate for each of the following conditions: control, 5-, or 80-fold MIC treatments). Each well was sheeted with 30 silicone coupons and incubated under static condition at 37 °C for 24 h to form biofilm on silicone coupons. Then, the exhausted medium containing floating cells was removed from the well, and the well was gently washed twice with 5 mL of LB medium as described above (hereafter, called washing step). Biofilms were incubated with 5 mL of fresh LB medium for an additional 24 h (in total of 48 h, Fig. 1a, step 1, on average this corresponded to ca $5 \times 10^6$ bacteria/coupon, ie $2.6 \times 10^5$ bacteria/mm²). After performing washing steps twice, one coupon was collected from each of the 9 wells in 500 μL of LB medium into a microtube (Fig. 1a step 2; hereafter, called coupon sampling). Next, biofilm cells contained in 3 wells were incubated with LB (control), with 5- or 80-fold MICs of AMK dissolved in 5 mL of LB medium (Fig. 1a step 3) under static condition at 37 °C for 24 h. After removing the supernatant containing AMK and performing washing steps twice (Fig. 1a step 4), one coupon was collected in each well (Fig. 1a step 5). Then, 5 mL of LB medium was poured into the well to allow survived biofilm cells to regrow. Next, the survived biofilm cells were incubated for 24 h at 37 °C, and coupon sampling was performed in each well after removing the supernatant and performing washing steps twice (Fig. 1a step 6). Finally, the biofilm cells were incubated with 5 mL of LB medium for an extra 24 h (in total of 48 h) before the subsequent AMK exposure (Fig. 1a step 2). Steps 2–6 were defined as a series of cycles, and the experiment is performed for 10 cycles. In each sampling step, bacterial suspensions were prepared by detaching biofilm cell from coupons as mentioned above. Then, the bacterial number of the suspension was confirmed by performing CFU counts and stocked as glycerol stock at −80 °C and characterized later. From CFU counts of last cycle at step 5, some colonies were randomly picked as endpoint clones and used for characterization. After 10 cycles of evolution the biofilm biomasses at step 2 before last treatment represented on average $8 \times 10^9$ bacteria/coupon with no treatment, $4 \times 10^9$ bacteria/coupon at 5xMIC and $6 \times 10^8$ bacteria/coupon at 80xMIC.

*Planktonic condition*. First, LF82_GFP and LF82_mars from glycerol stocks were grown for 24 h in LB medium at 37 °C for 24 h under shaking condition. The two cultures were adjusted to OD$_{600}$ of 0.01 in LB medium into test tubes and incubated for an extra 24 h (in total of 48 h, Fig. 1b step 1). LF82_GFP and LF82_mars cultures were mixed at a 1:1 ratio, after diluting each culture to OD$_{600}$ of 2.0. Then, 200 μL of the mixed culture was collected and stored for further analyses (Fig. 1b step 2). Next, 3 mL of the mixed culture was transferred into 9 test tubes and 3 cultures were incubated without AMK, 3 with AMK at 5-MICs and the remaining 3 cultures with AMK at 80-fold MICs (Fig. 1b step 3). After 24 h of incubation under shaking condition, 1 mL aliquots of the 9 cultures were centrifuged, and bacterial pellets were harvested, washed twice, and suspended in 1 mL of LB medium (Fig. 1b step 4). Then, 500 μL aliquots of the 9 suspensions were stored for further analyses (hereafter, called sampling step, Fig. 1b step 5). Next, 30 μL of each suspension was inoculated in 3 mL of fresh LB medium (i.e., 100-fold dilution) and incubated for 24 h at 37 °C under shaking condition. Subsequently, after performing bacterial collection washing- and sampling- steps (Fig. 1b step 6), 1/100-

fold suspension was inoculated in 3 mL of LB medium and incubated before the subsequent AMK treatment (Fig. 1b step 2). Steps 2–6 were defined as a series of cycles, and the experiment is performed for 10 cycles. The last cycle was stopped at step 5. In each collected culture, the bacterial number was confirmed by performing CFU counts and stocked as glycerol stock at −80 °C and characterized later. From CFU counts of last cycle at step 5, some colonies were randomly picked as endpoint clones and used for characterization.

**Evaluation of the numbers of generation propagated during experimental evolution experiments**. The numbers of generation during which bacteria were propagated were evaluated using CFU counting performed at step 2, 5 and 6 of each cycle of evolution, both for biofilm and planktonic populations (Fig. 1). For biofilms, generation time was calculated by calculating equation (1) $n1 = \log_2(\text{CFUstep6}) - \log_2(\text{CFUstep5})$ and equation (2) $n2 = \log_2(\text{CFUstep2}) - \log_2(\text{CFUstep6})$. For planktonic populations, because of two steps where 1/100-fold dilution was performed, generation time was calculated by calculating equation (3) $n1 = \log_2(\text{CFUstep6}) - \log_2(\text{CFUstep5}/100)$ and equation (4) $n2 = \log_2(\text{CFUstep2}) - \log_2(\text{CFUstep6}/100)$. Therefore, for each cycle the total number of generation times was $n1 + n2$. Calculated generation times were then cumulated over the cycles (supplementary Data 1). At the end of the evolution cycles populations have been propagated for ~77, ~200 and ~36 generations respectively for biofilms treated with 5xMIC, 80xMIC and non-treated, and for ~324, ~212 (after 3 cycles since then bacteria were fully eradicated) and ~250 generations respectively for planktonic populations treated with 5xMIC, 80xMIC and non-treated (supplementary Data 1).

**Frequency of amikacin resistant mutants**. Aliquots of frozen stocks of evolved biofilm- and planktonic- populations obtained from step 5 (Fig. 1a, b) in each cycle were directly inoculated to LB medium into test tubes and incubated overnight under shaking condition at 37 °C. As described above, CFU counts for these cultures were performed on LB agar plates with or without AMK at 1-, 2- or 4-fold MICs. The frequency of AMK resistant-mutants was calculated for each AMK concentration as follows equation (5):

$$frequency\ of\ AMK\ resistant\ mutants(\%) = \frac{CFU_{AMKat1\times,2\times or 4\times MICs}}{CFU_{LB}} \times 100(\%)$$

where $CFU_{AMKat1\times,2\times,or4\times MIC}$ is the CFU number of each population grown on LB agar plates containing AMK at 1-, 2- or 4-fold MICs, and $CFU_{LB}$ is the one grown on LB agar plate.

**Frequency of rifampicin resistant mutants in biofilm and planktonic cells**. As mentioned above, after forming 24 h LF82 biofilms on silicone coupons in LB medium in a well of a 6-well plate, a bacterial suspension of biofilm cells was prepared. For planktonic cells, after growing LF82 in a test tube in LB at 37 °C for 24 h, 2 mL aliquots of planktonic culture were harvested, washed, and suspended in 120 μL LB medium. Then, the bacterial numbers of each suspension from biofilm and planktonic conditions were confirmed by CFU counts on LB agar plates. Additionally, 100 μL of each bacterial suspension was plated on LB agar plates with RIF at 4-fold MIC (MIC of RIF against LF82 is 16 μg/mL). After 24 h incubation, CFUs were quantified from each plate. The frequency of RIF-mutants was calculated as the ratio of the bacterial number on LB agar plate with RIF at 4-fold MIC per that the one on LB agar plate without RIF.

**Survival of endpoint clones**
*Biofilm cells on silicone coupons*. Percentages of survival of endpoint clones from biofilm evolution on biofilms against 5- and 80-fold MIC of AMK were evaluated as described above. Briefly, the randomly picked clones were overnight cultured and diluted to an optical density at 600 nm (OD$_{600}$) of 0.05 in LB medium, and 5 mL of the diluted culture was poured into the well with silicone coupons. Biofilms were grown at 37 °C for 24 h under static condition. After 24-h incubation, the exhausted medium containing floating cells was removed from the well using an aspirator (VACUSIP; Integra Biosciences), and the well was gently washed once with 5 mL of PBS (Lonza, Rockland) to remove un-adherent floating bacteria. Three coupons were aseptically collected using sterilized forceps and each dipped in 500 μL of LB medium in a microtube as before treatment coupons. The remaining biofilm cells in the well were treated with AMK dissolved in 5 mL of fresh LB medium at 37 °C for 24 h. After removing the supernatant, the well was rewashed, and three coupons were collected each in 500 μL of LB medium as treatment coupons. Each microtube containing one coupon (before treatment coupons and treatment coupons) was vigorously vortexed for 1 min and sonicated for 10 min using an ultrasonic bath (Branson 5800, Branson Ultrasonics, CT, USA) to detach biofilm cells from coupons and prepare bacterial suspension. The bacterial number of the suspension was confirmed by performing CFU counts.

*Planktonic cells*. Percentages of survival of endpoint clones from biofilm and planktonic cell evolution of planktonic cells against 5- and 80-fold MIC of AMK were evaluated as described above. Briefly, strains were incubated in a test tube containing 3 mL of fresh LB medium at 37 °C for 24 h under shaking condition. After 24 h incubation, LF82 culture was treated with AMK at 37 °C for 24 h under shaking condition by directly adding 5- or 80-fold MIC of AMK in the test tube.

Before and after the AMK treatment, 200 µL of the culture was collected in a microtube. Then, bacterial pellets were harvested by centrifuging at $15,000 \times g$ for 3 min at 25 °C, washed once with 200 µL of PBS, and suspended in 200 µL of LB medium to prepare bacterial suspension. The bacterial number in the suspension was confirmed by performing CFU counts. The test was repeated six times for each concentration.

**Genome sequencing and analysis**. The genomic DNAs were extracted from LF82 mutants (generated constructs and end-point clones of evolution experiments) and evolved populations obtained from experimental evolution. The glycerol stock was scraped, incubated in 1 mL of LB and incubated for 16 h at 37 °C under shaking conditions. The cells were then pelleted and used for genome extraction using the Wizard Genomic DNA Purification kit (Promega, Madison, WI, USA) and Qia-quick PCR purification kit (Qiagen, Hilden, Germany). The extracted genomic DNAs were prepared for whole genome sequencing using the Nextera XT DNA library preparation kit (Illumina, San Diego, CA, USA) and sequenced on Hiseq and Miseq platform (Illumina) with an average depth of 150x. Reads quality was assessed using FastQC version 0.11.9 (http://www.bioinformatics.babraham.ac.uk/projects/fastqc/) and trimmed using Trimmomatic version 0.39[102]. Trimmed sequence reads were analysed by BreSeq version 0.35.0[103] to detect genetic variants using default parameters for clones and by adding the -p flag for the populations. Only mutations at higher frequency than 5% are detected by BreSeq. No chromosomal rearrangements or movement of insertion elements were detected as selected by the applied treatment. The Muller Plots were inferred and produced using the lollipop tool version 0.9.0 (https://github.com/cdeitrick/Lolipop) as well as the ggmuller R package 0.5.3[104].

**Competition assay for *fusA* and *sbmA* mutants in the presence or absence of AMK**. Competition assay for *fusA* and *sbmA* mutants were performed according to a previously published method[10], with some modifications. Briefly, LF82_mars, LF82_GFP, and evolved strains (*fusA* and *sbmA* mutants) tagged with *mars* were grown overnight in LB at 37 °C. All overnight cultures were diluted to OD$_{600}$ of 2.0. Then, LF82_mars and mars-tagged evolved strains were mixed with LF82_GFP at a 1:1 ratio. The mixture of LF82_mars and LF82_GFP was used as a reference competition. Before starting competition assays, each mixture was verified to be in 1:1 ratio by MACSQuant VYB flow cytometer (Miltenyi Biotec, Bergisch Gladbach, Germany). Each mixture was diluted 300-fold with PBS in a microtube. Then, 200 µL of each diluted sample was inoculated in a 96-well plate for the flow cytometry analysis. At least 10,000 events per sample were acquired. An abundance of cells gated as 'red- positive' or 'green-positive' depended on competition combinations. The gating strategy is explained in Supplementary Fig. 14. The software used for analysis was MACSQuantify™ (Miltenyi Biotec).

The experiments (i) and (ii) were performed using 3 biological replicates in each competition.

The relative fitness of each evolved strain was calculated as follows equation (6):

$$relative\ fitness = \frac{\left(\frac{evolved_{red}}{wt_{green}}\right)}{\left(\frac{ref(wt)_{red}}{ref(wt)_{green}}\right)}$$

where $wt_{green}$ and $evolved_{red}$ are the cell proportions of the LF82_GFP and mars-tagged evolved strains in the competition mixture, and $ref(wt)_{green}$ and $ref(wt)_{red}$ are the ones of LF82_GFP and LF82_mars in the reference competition mixture, respectively.

*Competition assay without AMK treatment*. Each OD$_{600}$ 2.0 adjusted mixture was diluted to OD$_{600}$ = 0.02 in 3 mL of LB medium and incubated 24 h at 37 °C under shaking condition at 180 *rpm*. The relative amount of green and red cells was analysed using flow cytometer to calculate the fitness cost for each evolved strain in the absence of AMK treatment.

*Competition assay with AMK treatment*. Three mL of each mixture at OD$_{600}$ of 2.0 was treated with 5-fold MIC AMK for 24 h at 37 °C under shaking condition at 180 *rpm*. Then, 1 mL of each mixed culture was collected in a microtube. Next, bacterial pellets were harvested by centrifuging at $15,000 \times g$ for 3 min at 25 °C, washed twice with 1 mL of LB, and suspended in 1 mL of fresh LB medium to prepare bacterial suspension. Next, 30 µL of the suspension was inoculated in 3 mL of fresh LB medium and incubated 24 h at 37 °C under shaking condition. Finally, the relative amount of green and red cells was analysed using flow cytometer to calculate the fitness cost for each evolved strain in the presence of AMK treatment.

**Bacterial number and biofilm amount of *fimH* mutants formed on silicone coupons**. Overnight cultures of LF82_mars and *fimH* mutants grown in LB at 37 °C were diluted to OD$_{600}$ of 0.05. Five mL of the diluted culture was inoculated in a well containing 24 silicone coupons and incubated for 3 h at 37 °C under static condition. Then, the supernatant was removed from the well, and the well was gently washed with 5 mL of LB medium once to remove un-adherent bacteria. Next, 5 mL of fresh LB medium was newly added to the well and incubated for 21 h. Then, the exhausted medium containing floating cells was removed, and the

well was washed with 5 mL of LB medium twice. Experiments (i) and (ii) were performed using 3 biological replicates in each strain.

*CFU counts of biofilm cells on silicone coupons*. A total of 6 coupons were randomly collected in 3 microtubes containing 500 µL of LB medium (i.e., 2 coupons/microtube). Bacterial suspensions were prepared to perform CFU counts as described above.

*Cristal violet assay to quantify biofilm amount on silicone coupons*. After collecting the 6 coupons for CFU counts, 6 mL of 1% crystal violet (CV) solution (Sigma-Aldrich) was added to the well and incubated for 15 min at 25 °C. The well stained with CV was gently washed with 7 mL of PBS three times to remove excess CV, and the plate was dried up for 1 day in a chemical hood. On the next day, a total of 18 coupons were randomly collected in 6 microtubes (i.e., 3 coupons/microtube), which contains 810 µL of the mixed solution of ethanol/acetone at 80%:20% ratio. Coupons were suspended in the solution for 15 min to allow the CV stain to be dissolved. After transferring 100 µL of dissolved CV solution in a 96-well plate, the OD$_{570}$ value was measured to quantify the CV solution using a multimode plate reader (Tecan Infinite M200 PRO). As the background control, 6 coupons sheeted in a well containing 5 mL of LB medium without bacteria were also treated similarly.

**Survival rate of biofilm cells of *fimH* mutants formed on silicone coupons after AMK treatment**. Similar to the above, 5 mL of the diluted culture at OD$_{600}$ of 0.05 was inoculated in the well containing 8 coupons and incubated for 3 h at 37 °C. After washing the well once, 5 mL of fresh LB medium was newly added in the well and incubated for 21 h at 37 °C. Then, after washing the well twice, 4 coupons were collected in one microtube containing 500 µL of LB medium (i.e., 4 coupons/microtube). Then, 5 mL of LB medium containing AMK at 80-fold MIC was added to the well and incubated for 24 h at 37 °C. Next, after washing the well twice to remove AMK, the remaining 4 coupons were collected in one microtube containing 500 µL of LB medium (i.e., 4 coupons/microtube). Bacterial suspensions were prepared from microtubes containing coupons, as mentioned above. Subsequently, CFUs were quantified on LB agar plates in the same way as described above. Finally, the survival rate of each strain was calculated as follows equation (7):

$$\%\ survival = \frac{(CFU_{aftertreatment}/coupon)}{(CFU_{beforetreatment}/coupon)} \times 100(\%),$$

where $CFU_{aftertreatment}$ is the bacterial number of the suspension for 4 coupons collected from the well after AMK treatment, and $CFU_{beforetreatment}$ is the one for 4 coupons collected from the same well before AMK treatment, respectively. This experiment was performed using 9 biological replicates for each strain.

**Bioinformatic analysis of FusA sequences**. All *E. coli* FusA proteins contained in the nrprot database of NCBI (version 2019-04-09) were identified by BLASTp (blast+ version 2.2.31[105]) and only hits with an e-value lower than $1 \times 10^{-10}$ and sequence identity >70% were kept. The resulting 2143 sequences were aligned using mafft version 7.407[106] with G-INS-I option and the alignment was used to compute entropy in each structural domain of the protein using the HIV sequence database website tool (https://www.hiv.lanl.gov/content/sequence/ENTROPY/entropy.html).

The same alignment was also screened to identify mutations relatively to our reference sequence from LF82 as in ref. [50].

**Growth capacity**. An overnight culture of each strain was diluted to OD$_{600}$ of 0.05 in fresh LB medium. Two hundred-µL aliquots were inoculated in a 96-well plate. The plates were then incubated in a TECAN Infinite M200 Pro spectrophotometer (Männedorf, Switzerland) for 20 h at 37 °C with shaking of 2 mm amplitude. The absorbance of each culture at 600 nm was measured every 15 min. Growth curve of each strain was measured using 3 biological replicates.

The R package GrowthCurver[107] was used to infer the growing capacities for all strains based on the growth curves.

**Penetration of antibiotics through biofilms**. Penetration of antibiotics through biofilms were determined according to a previous report[108] with slight modifications. In brief, the *E. coli* LF82 cultures were grown overnight in LB at 37 °C with shaking and then diluted in the same medium to an optical density of 0.05 at 600 nm with PBS. A 10 µL drop of the diluted culture was used to seed black, polycarbonate membranes (diameter, 13 mm; pore size, 0.4 mm) (Millipore US) placed on TSA plates. The plates were inverted and incubated at 37 °C for 48 h, with the membrane-supported biofilms transferred to fresh culture medium every 24 h. The membrane-supported biofilms were transferred to MHA plates inoculated with *S. aureus* ATCC 29213, a quality control strain of *S. aureus* set to McFarland standard 0.5, so as to give a confluent lawn of growth after incubation. A 6 mm nitrocellulose membrane (pore size, 0.4 mm; Millipore) was then placed on the surface of each biofilm along with an antibiotic disc (oxacillin, 1 mg; cefotaxime, 30 mg; amikacin, 30 mg; ciprofloxacin, 5 mg and vancomycin, 30 mg;

all from Becton Dickinson, US) pre-moistened with 24 µL of sterile distilled water placed on top of it. Control assemblies consisting of sterile membranes and anti-biotic discs, without biofilm, were set up in parallel. The plates were incubated for 24 h at 37 C and the zones of growth inhibition on MHA plates with test and control assemblies were measured. Experiments were performed in triplicate. Two-tailed, paired *t*-test was used for statistical analysis.

**Statistics and reproducibility**. Data are presented as the mean ± standard deviation (SD) or ± standard error of the mean (SEM) or with individual data points from at least three independent biological experiments. Statistical ana-lyses were performed using Prism 9.5.0 (GraphPad Software Inc.) and corre-spond to unpaired two-tailed *t*-test with Welch's correction between two groups and to oneway analysis of variance (ANOVA) followed by TukeyHSD post-hoc test for more than two groups. *P*-values <0.05 were defined as the level of statistical significance. FACS data were analysed by the software MACSQuan-tify™ (Miltenyi Biotec).

**Reporting summary**. Further information on research design is available in the Nature Portfolio Reporting Summary linked to this article.

## Data availability
All data of this study are included in this published article and in its supplementary information. All sequencing reads were deposited in NCBI under the BioProject accession number PRJNA833264. All source data underlying the graphs presented in the main figures are available as supplementary Data 8.

## Code availability
The codes used in this study are available at https://github.com/Sthiriet-rupert/Usui_Ecoli_Amk.

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

## Acknowledgements

We thank Dr. Olaya Rendueles, Dr David Lebeaux and Dr Rebecca Stevick for critical reading of the manuscript. We are grateful to Dr. Olaya Rendueles for the initial help with the analysis of the mutations. This work was supported by the French National Research Agency (ANR), project EvolTolAB (ANR-18-CE13-0010), by the French government's Investissement d'Avenir Program, Laboratoire d'Excellence "Integrative Biology of Emerging Infectious Diseases" (grant n°ANR-10-LABX-62-IBEID) and by the Fondation pour la Recherche Médicale (grant DEQ20180339185). S.T.-R was supported by the French National Research Agency (ANR), project EvolTolAB (ANR-18-CE13-0010).

## Author contributions

C.B. and M.U. designed the experiments. M.U., Y.Y. and S.T.-R. performed the experiments. C.B., M.U., Y.Y, S.T.-R. and J.-M.G. analysed data. C.B. wrote the manuscript with significant help of M.U., Y.Y, S.T.-R. and J.-M.G.

## Competing interests

The authors declare no competing interests.
