## [Peer Review File · Communications Biology]

Reviewers' comments:

Reviewer #1 (Remarks to the Author):

The manuscript by Usui et al presents an interesting study showing mutations in the *E. coli* elongation factor G (*fusA*) and an inner-membrane transporter (*sbmA*) that were not only selected for, but also preferentially retained during experimental evolution of antibiotic resistance to the aminoglycoside amikacin in biofilms but not during planktonic growth. This study advances our understanding of the complex nature of evolution to antibiotic resistance, specifically, within biofilms. The manuscript is technically well written but at multiple instances hard to follow (highlighted below) and overall appears very long. There is an abundance of supplementary experiments performed to support the claims made within the manuscript, all of which highlight the importance of understanding how bacteria evolve antibiotic resistance during biofilm-based infections. But this extended information at certain points distracts the reader from the main story.

Major questions and issues:

Ln35-52 (abstract) : The abstract is vague in its details, improvements could be made including – e.g., detailing what *sbmA*, *fusA*, and *fimH* encode for. This would help with following the manuscript in the early stages. The transition to *fimH* is not clear and confusing (Ln48) and the concluding statement is not of much relevance and distracting (Ln50-52).

The examples given in Ln93-104, together with the findings presented in this study, make me wonder how robust those biofilm experiments were? How much of a 'real' biofilm does in vitro growth on a coupon in nutrient rich media represent? How thick are the biofilms on coupons – in other words, could those discrepancies among all those identified studies (including the one presented here) be explained by monolayer vs matured biofilms? The authors mention an 'initial drop' (Ln149) in bacterial numbers in their biofilm experiments, which could be indicative that most of the bacteria were not in a proper biofilm yet. Also, based on Fig 8 one could argue that this *E. coli* strain is a poor biofilm former especially under the investigated conditions.

Considering the importance of the antibiotic amikacin in this study, the introduction should further explain the specific mode of action of this antibiotic. What is known about its properties to diffuse through a biofilm? Since aminoglycosides are protein synthesis inhibitors, I wonder why these biofilms were highly metabolically active (could this potentially be due to a monolayer rather than a mature biofilm)? Based on the EUCAST classification (MIC breakpoint of 8 ug/ml for Enterobacterales), the used *E. coli* strain appears amikacin resistant– how does this affect the overall interpretation of their findings? What is known in the literature about *E. coli* developing resistance to amikacin? How quickly (or not) does this happen? Their planktonic data is very interesting, but I wonder how this aligns with the wider literature?

The claim that amikacin resistance occurred faster in biofilm than in planktonic conditions is troublesome in my opinion as different antibiotic concentrations were used and no data on how much antibiotic actually reaches the bacterial cells in the biofilm exists (esp. what about the bacterial cells that are deeply attached / metabolically inactive etc.). Also, important information regarding the potency of amikacin against biofilms is missing - What are the minimum biofilm inhibitory and eradication concentrations? What is the CFU count in a biofilm vs the planktonic cells? Please provide a justification for the use of 5x and 80x MIC values for the experimental evolution experiments.

Figure 2 and supplementary figures: Is the y-axis on this plot log normalized and also normalized relative to survival CFUs of no treatment? The overall representation of 'percentage survival' is confusing and difficult to understand as to what the exact amount of difference between planktonic and biofilm survival is – why not show the real CFU values that were obtained? The methodology how bacterial cells were washed seem problematic (Ln582) – the high centrifugation speed has the

potential to damage the bacterial cells which could alter the CFU counts.

What do the authors speculate why there were no mutations in their planktonic cultures considering that A) their culture model let bacteria grow into late stationary phase (nutrient starvation) and B) using glass tubes with potential lower oxygen exposure (oxygen stress)? With the experimental setup alone, one would expect more mutations.

Ln220-222: were any of those mutations confirmed with individual created knockout mutants? How many mutations are overall on the chromosome in these populations? It appears very descriptive without confirmation studies. How did the authors rule out that nothing else caused increased resistance in biofilms?

Regarding the early selection of *sbmA* and *fusA* mutations in biofilm – how relevant would this be in an established (thick) biofilm in a patient?

Other minor comments:

Ln60: experiments

Ln63-64 / Ln117: Mutations that lead to increased tolerance need to be further explained as it is not clear what the authors refer to. If cells 'just' survive, how can this lead to biofilms have a 'high mutation rate' as stated in the abstract?

Ln65: "before to favor" = "before favoring"

Ln65: remove "genetic"

Ln66: add 'a' before clinical setting"

Ln68: plural - infections

Ln71-74, 79-80: Based on that thought, traditionally, those bacterial species isolated from biofilm-associated infections were antibiotic susceptible in the lab – so how does this contribute to the overall 'selling point' of this manuscript?

Ln81-83: this should be further explained with clear examples

Ln90: "biofilm high antibiotic tolerance" change to "biofilm associated high antibiotic tolerance"

Ln92: "speed" = "rate"

Ln94: remove "colony"

Ln104-106: This sentence needs to be rephrased to mention the comparison of biofilm to planktonic growth as there is no doubt that biofilms "constitute a dynamic evolutionary reservoir enabling the emergence of high antibiotic resistance"

Ln75-106: overall this paragraph sound more like a discussion than introduction

Ln123: can the authors reference the clinical relevance accordingly?

Ln143: remove "as"

Ln144-148: What does this mean and what are the authors trying to communicate?

Fig 2(C/D): I am unsure if showing the MIC mean and standard errors of the mean for 3 independently evolved lines is appropriate. How many times was the MIC determined for each evolved line at each step? Was there variability within this number? Additionally treating these independently evolved lines as if they are a replicate set is slightly misleading, perhaps showing each independent line MIC value at each cycle would be more appropriate.

Fig 3, Fig S6, Table S2: What was the rationale of selected genes (also gene names must be italicized) and how many samples were used for frequency calculations? Moreover, although it is clearly mentioned that planktonic lines 4-6 (exposed to 80x MIC) did not survive the entire 10 selections, it is stated that those populations were sequenced at their last survival stage. Where is the data for this?

In Table S2 it states they were not determined, is this correct, or was there not detectable mutations in the populations above 5% frequency? Regarding the colouring of these figures: It is very difficult to distinguish mutations which occur rarely from those which do not occur at all (eg: *yhdP* in control line 9, in figure 3). I would suggest going from one colour to another (not white to red), or perhaps more appropriately putting a % value on each square with a value above 5% so the reader can clearly distinguish boxes which have for example 20% mutation frequency and those which have 50% (much like is done for Figure 5).

Ln168-169: Using the word "evolution" when talking about increases in MIC confuses the terminology, as the MIC itself is not evolving it is symptom of the bacteria evolving resistance, I would stick with just using terms like increase.

Ln172-177: The use of the word "clones" here is inconsistent with the terminology being used in the figures either rephrase to call them "resistant clones" or "resistant mutants".

Ln188-189: Although an accurate statement in the context of the whole manuscript -It is not until you sequence individual clones (figure 5) that you could have been certain that the survival you were observing was caused by resistance and not a combination of resistance and tolerance, perhaps this sentence is premature in the manuscript.

Ln208-211: FimH mutations are also seen in planktonic cultures without antibiotics

Figure 4: gene names must be italicized

Ln410: Incorrect supplementary figure referred to – should be S12.

Ln416-429: this discussion is mainly speculation and does not add much; could be deleted

Fig 8: the increased biofilm formation is questionable based on those low numbers and large overlap (the CFU counts are difficult to understand as the y-axis scale is not appropriate for this data).

Ln431: It should be noted also that mutations in *fusA1* in *P. aeruginosa* have been shown to confer resistance to aminoglycosides (including amikacin) alongside being commonly selected for during infection (see Bolard et al, 2018. 10.1128/AAC.01835-17).

Ln475: Are the high rate of *fimH* mutations (leading to enhanced biofilm formation) not evidence of increased antibiotic tolerance mechanisms alongside antibiotic resistance mechanisms contributing to survival and resistance?

Ln582: please state x g not rpm

Ln590: please state x g not rpm.

Ln702: remove space between oC

Ln723: please state x g not rpm.

Ln737: How were sequences trimmed, were they quality and adapter trimmed, using which software?

Ln741: please add version of lollipop used (if one exists)

Ln742: please correctly cite ggmler (see the readme for ggmler on instructions for this - <https://cran.r-project.org/web/packages/ggmler/readme/README.html>)

Ln789: please state x g not rpm.

Ln856: Thank you for making all code used available, please ensure before publication the BioProject is released.

Ln877: Please ensure consistency across references included, some contain DOIs and other do not.

Fig S9: How many times/replicates were done for each growth curve.

Reviewer #2 (Remarks to the Author):

Masaru et al. carried out a tour de force study comparing evolution of amikacin resistance for pathogenic *E. coli* LF82 grown under biofilm or planktonic conditions when treated with intermittent lethal levels of antibiotic. Masaru et al. observed bacteria accumulated more mutations, mainly in *sbmA* and *fimH* genes, when grown in a biofilm compared to planktonic growth conditions. In particular, the authors notice that *fimH* is rarely observed for bacteria when grown under planktonic conditions, which the authors attribute to imparting a fitness cost for bacteria grown under planktonic conditions. Overall, this is an impressive amount of work with high quality data. However, it can be difficult to follow sometimes due to technical details and phrasing, and as a result, may be more appropriate for a specialized journal.

General comments:

It is still not clear why bacteria grown in a biofilm accrue more mutations compared to bacteria grown under planktonic conditions. The authors should carry out sequencing studies to confirm this by growing bacteria under the two conditions, but without antibiotics.

The authors place a lot of emphasis of phenotypes observed when a sbmA mutations are not present. However, there are other mutations present that could also explain the phenotype.

Specific comments:

Line 104: 'Hence, whether or not antibiotic-tolerant biofilms constitute a dynamic evolutionary reservoir enabling the emergence of high antibiotic resistance is still unclear'

Based on such different results observed in the literature, it seems like evolution of resistance is very context dependence and maybe there isn't a universal explanation (i.e., biofilm always evolves resistance faster or at greater diversity compared to planktonic, or vice versa).

Line 116: please list what are the typical mutations observed.

Line 128: please list proposed alternative treatment options (i.e., different treatment regimes, peptides, phages, exc.).

Line 142: 'superior to the mutation prevention concentration (MPC: 64 µg/mL)': how was this concentration determined. You are clearly observing mutations enabling resistance so it is not mutation prevention?

Figure 1: incomplete data for planktonic 80x. The authors mention doing ~200 plus cycles, but only show data for 3 cycles.

Figure S3 should be moved into the main text. It is very informative having the raw CFU counts instead of %. It rules out the possibility that there are simply more bacteria for biofilm conditions, and that is why more mutations are observed.

Figure 3: Why are the replicates labeled 1-9? Labeling it as replicate 1-3, 1-3, 1-3 would be clearer. Also, frequency should be on the scale of 0-1.

Line 210: 'Although these mutations are not directly associated to antibiotic resistance, they might increase biofilm formation and have an impact on survival against antibiotics (see below).' Please continue the discussion, rather than having to fish for it later. Or add, see below in section..'

Line 216: It isn't clear what 'sample 3' is referring to (i.e., Figure 3 or Figure 4). If Figure 3, could the explanation be the presence of another mutation (fadE) rather than the absence of a sbmA mutation not leading to enhanced resistance? If referring to Figure 4, there is no sample 3 for planktonic condition.

Line 230: How was this determined, by increased MIC or literature precedent: 'enhanced genetic antibiotic resistance – and not tolerance –' ? If by MIC, please include rational in sentence.

Reviewer #3 (Remarks to the Author):

Usui et. al present a study investigating the evolutionary trajectory of amikacin resistance in planktonic and biofilm cultures. Evidently, the study of resistance evolution is of great clinical concern and the insights gained may benefit improved antibiotic stewardship. Using parallel sequential passaging with intermittent antibiotic exposure, the group finds that the biofilm state selects for more rapid and potent resistance mutations than in planktonic cultures, suggesting that the nature of infection can impact the emergence of resistant clones. Altogether, the manuscript presents a compelling investigation of bacterial evolution of antibiotic resistance. However, the work could benefit from additional experiments that establish a stronger causal link between specific mutations and fitness/resistance outcomes. These additional experiments would add more biological insight into the findings. Nonetheless, the observation of differential evolutionary trajectories of resistance mutations in different cellular lifestyles is compelling.

General Comments

- The quality of writing is generally quite good, but there are typos and misuse of word choice throughout that detract from the understandability of the article. I would recommend further editing, particularly in the introduction.

o Ex: Line 38: "much less is known about the development [...]; Line 41-42: "relevants cycles of lethal treatment"; Line 59-60: 'The use of adaptive laboratory experiments'. Etc

- The hypothesis that the biofilm state selects for different and more potent amikacin mutations as planktonic growth is supported by the evolution experiments. However, I believe these observations would be significantly strengthened if a third sequential passaging experiment was performed wherein the cultures were made to oscillate between the coupon-biofilm growth and planktonic growth. In that case, one could observe if the cost of the resistance mutations in planktonic growth outcompetes the benefits of resistance that arises during biofilm growth.

- A major concern of the paper are the biological conclusions drawn from the whole genome sequencing of individual clones from the passaging experiments. While it's clear that the sbmA, fusA, and fimH mutation likely impact amikacin resistance, it is difficult to demonstrate causality without comparison of these mutations in otherwise isogenic strains. Based on the results in Figure 7, it seems like the group has the capacity to genetically modify the E. coli strains and perform the MIC/competition experiments with interesting fusA and sbmA mutations. With these experiments, much stronger and more informative claims can be made on the sufficiency of these mutations to confer increased resistance or biofilm tolerance or if these mutations are only relevant in the context of additional mutations in the genome.

Specific Comments

- Line 198-199: SbmA is described as an inner membrane transporter associated with increased resistance to aminoglycosides. One would expect loss-of-function mutations to sensitize strains to amikacin, but it seems like the opposite is observed in the experiments. Please add some additional discussion to better clarify the purported nature of these mutations.

- The discussion of the contents of Fig. 7 in the Results are inadequate. Are these naturally occurring mutants from the evolution experiments? Or have they been generated through genetic modification? Is the truncated mutant expected to phenocopy the point mutation? Please expand the discussion of results.

Reviewer #1 (Remarks to the Author):

The manuscript by Usui et al presents an interesting study showing mutations in the *E. coli* elongation factor G (*fusA*) and an inner-membrane transporter (*sbmA*) that were not only selected for, but also preferentially retained during experimental evolution of antibiotic resistance to the aminoglycoside amikacin in biofilms but not during planktonic growth. This study advances our understanding of the complex nature of evolution to antibiotic resistance, specifically, within biofilms. The manuscript is technically well written but at multiple instances hard to follow (highlighted below) and overall appears very long. There is an abundance of supplementary experiments performed to support the claims made within the manuscript, all of which highlight the importance of understanding how bacteria evolve antibiotic resistance during biofilm-based infections. But this extended information at certain points distracts the reader from the main story.

Answer:

Thank you for these comments. We agree that the text could be further clarified and shortened and we have extensively revised and proofread the manuscript to facilitate the reading. All modifications are highlighted in yellow.

Major questions and issues:

Ln35-52 (abstract) : The abstract is vague in its details, improvements could be made including – e.g., detailing what *sbmA*, *fusA*, and *fimH* encode for. This would help with following the manuscript in the early stages. The transition to *fimH* is not clear and confusing (Ln48) and the concluding statement is not of much relevance and distracting (Ln50-52).

Answer:

We modified the abstract according to your comments.

The examples given in Ln93-104, together with the findings presented in this study, make me wonder how robust those biofilm experiments were? How much of a 'real' biofilm does *in vitro* growth on a coupon in nutrient rich media represent? How thick are the biofilms on coupons – in other words, could those discrepancies among all those identified studies (including the one presented here) be explained by monolayer vs matured biofilms? The authors mention an 'initial drop' (Ln149) in bacterial numbers in their biofilm experiments, which could be indicative that most of the bacteria were not in a proper biofilm yet.

Answer:

The reviewer has a good point and some of the differences among antibiotic evolution experiments could be partly due to different levels of biofilm maturity, in particular when using different bacteria and different set-ups. However, in our experiments, the average CFU of the initial 48h biofilms that were subjected to evolution was ca. $5 \cdot 10^6$ CFU/coupon (as can be seen on the top panel of supplementary Fig. S3) and increased over the cycles of evolution to reach ca 10^9 CFU/coupon at the end of the evolution experiments ($8 \cdot 10^9$ CFU/coupon with no treatment, $4 \cdot 10^9$ CFU/coupon at 5xMIC and $6 \cdot 10^8$ CFU/coupon at 80xMIC).

The surface of a coupon is 19.625 mm² resulting in $2.6 \cdot 10^5$ bacteria/mm² in the initial 48h biofilms ($5 \cdot 10^7$ bacteria/mm² on average at the end of the experiments). This value corresponds to a high and consistent biofilm biomass on the coupons.

This number of bacteria is equivalent or higher than the ones obtained from biofilms used in evolution experiments in Santos Lopez et al. 2019, (10^4 to 10^5 *A. baumannii* per mm²); in Trampari 2021 (10^4 *S. Typhimurium* bacteria per mm²) and in Scribner et al 2020 ($6.5 \cdot 10^4$ *A. baumannii* and $6.5 \cdot 10^5$ *P. aeruginosa* per mm²).

In addition, the quantity of cells within biofilms increased overtime, and there is a clear difference in adaptive behavior to antibiotic treatment compared to what is observed for planktonic cells.

One should also keep in mind that during *in vivo* biofilm-related infections, high biomass, mature biofilms might not be so prominent, with biofilms mainly represented by relatively small bacterial aggregates (see Bjarnsholt, T. *et al.* The *in vivo* biofilm. *Trends in Microbiology* **21**, 466–474 (2013) and Sauer, K. *et al.* The biofilm life cycle: expanding the conceptual model of biofilm formation. 1–13 (2022). doi:10.1038/s41579-022-00767-0)

The maturity level of the initial biofilms used in our study is now indicated in the text.

line 141 : “We showed that biofilms treated with such periodic amikacin treatments displayed enhanced survival compared to planktonic populations (supplementary Fig. S1). However, we observed that the percentage of surviving biofilm bacteria showed an initial drop following the first cycle of 24h treatment, suggesting a relatively low level of biofilm maturity and tolerance.” and biomass quantification information are now indicated in the Material and Methods section line 547 and 564.

Also, based on Fig 8 one could argue that this E. coli strain is a poor biofilm former especially under the investigated conditions.

Answer:

Concerning the LF82 biofilm forming capacity: we use the *E. coli* strain TG1 carrying the F conjugative plasmid, a factor strongly promoting biofilm formation (Ghigo, *Nature* (2001), as a biofilm-positive control. When we compared the biofilm formation on silicon coupons by LF82 and TG1, the CFU/coupon for LF82 was ca $5 \cdot 10^6$ CFU/coupon and $2 \cdot 10^7$ CFU/coupon for TG1. Therefore, although the CFU/coupon of LF82 is lower than those of TG1, LF82 cannot be considered as a poor biofilm former.

Considering the importance of the antibiotic amikacin in this study, the introduction should further explain the specific mode of action of this antibiotic.

Answer:

We have now added the mechanism of action of amikacin in the introduction section:

Line 109:

“Amikacin is a bactericidal antibiotic that binds to bacterial ribosomes, inhibits protein synthesis and causes membrane disruption through production of aberrant proteins⁴⁹”.

What is known about amikacin properties to diffuse through a biofilm?

Answer:

Amikacin was described to diffuse through Staphylococcal biofilms (Singh R, *et al.* 2010. doi.org/10.1093/jac/dkq257) but its penetration is reduced in *P. aeruginosa* biofilms (Shigeta M *et al.* doi: 10.1159/000239587). We did not identify a study specifically investigating the penetration capacity of amikacin through *E. coli* biofilms and we agree, if this is what the reviewer had in mind, that a reduced capacity of amikacin to penetrate through *E. coli* biofilms could be involved in the observed enhanced tolerance of biofilms, among other possible mechanisms.

To further address this reviewer's comment, we have performed an additional experiment to evaluate whether various antibiotics could penetrate through LF82 biofilms. Our results are now shown as a new supplementary figure and demonstrate that all tested antibiotics, including amikacin, were capable of penetration through LF82 biofilms, but with a reduced efficacy.

This is now indicated in the discussion section (see figure below):

line 386: "This observed intrinsic tolerance of biofilms could also be partly explained by the possible reduced penetration of amikacin into biofilms as has been shown for *P. aeruginosa* biofilms⁹² and as we showed for *E. coli* LF82 colony biofilms (supplementary Fig. S12)."

Supplementary Fig. S12. Reduced penetration of antibiotics through LF82 biofilms. (A) Representative images of the penetration of antibiotics through the control assemblies and colony biofilms of *E. coli* LF82 as determined by the zone of *S. aureus* growth inhibition on MHA plates. (B) Averaged measurement of inhibition zones over triplicate experiments. The zones of growth inhibition (mean±SD) on MHA plates with control assemblies were taken as representative of 100% penetration and used to determine the percentage retardation of the penetration of antibiotics through biofilms. The limit of detection was 13 mm (size of the black, polycarbonate membrane filter on which biofilms were grown). S, significant. Two-tailed, paired *t*-test was used for statistical analysis.

Since aminoglycosides are protein synthesis inhibitors, I wonder why these biofilms were highly metabolically active (could this potentially be due to a monolayer rather than a mature biofilm)? Based on the EUCAST classification (MIC breakpoint of 8 ug/ml for Enterobacterales), the used *E. coli* strain appears amikacin resistant– how does this affect the overall interpretation of their findings?

Answer:

Thanks for pointing out this characteristic of the *E. coli* LF82 strain. Indeed, LF82 shows a tendency for enhanced resistance to amikacin as compared to *E. coli* K12. However, for the sake of coherence with the growing conditions that we used during our evolution experiments the LF82 amikacin MIC of 16 µg/mL corresponds to measurement made in LB (K12 strains have an amikacin MIC of 8 µg/mL in LB). We measured also the MIC of LF82 in MHA medium and it is 4 µg/mL (K12 strains have an amikacin MIC of 2 µg/mL in MH agar). Therefore, LF82 cannot be considered resistant based on EUCAST classification (MIC breakpoint of 8 µg/mL).

We have added a comment on this aspect line 510:

"The MIC of amikacin for LF82 in LB was measured as 16 µg/mL while its MBC was measured at 20 µg/mL. The MIC of amikacin for LF82 using Mueller Hinton medium was determined as 4

µg/mL, while K12 strains such as MG1655 had a MIC of 2 µg/mL. LF82 is therefore not resistant to amikacin based on EUCAST classification (MIC breakpoint of 8 µg/mL in MH)¹¹”.

Independently of LF82 MIC for amikacin, biofilm bacteria rapidly acquired enhanced level of resistance when exposed to intermittent treatment. This phenomenon did not occur in planktonic conditions, confirming the effect of the biofilm life style on the accelerated evolution towards the selection of antibiotic resistance. Interestingly, the resistance mutations to amikacin that we observed with LF82 have also been identified with some other different *E. coli* strains (see below), which therefore indicates that these mutations are not specific to the LF82 strain.

What is known in the literature about E. coli developing resistance to amikacin? How quickly (or not) does this happen?

Answer:

Many studies evidenced the evolution of aminoglycoside resistance (including to amikacin) in *E. coli*. In our manuscript, we cited studies (55 to 59 in the revised version of the manuscript), in which convergent mutations, notably *sbmA*, *fusA* or *cpxA*, have been systematically obtained when evolving *E. coli* in presence of amikacin (see line 197 and 205). Additionally, several other mutations in genes identified in our study were also shown to confer enhanced resistance to amikacin, such as the *fre* NAD(P)H flavin reductase encoding gene (ref 55 to 59), the *nuo* genes related to NADH-quinone oxidoreductase (ref 11 to 14 in the supplementary material).

This has now been updated in the revised version of our manuscript.

Evolution studies documenting amikacin resistance development in *E. coli* were generally performed using either gradient or incremental concentration of amikacin (starting with sub-MIC) (and not intermittent treatment with lethal concentration as we did here) (ref 55 to 57). Unfortunately, in these studies the mutational analyses have been performed at the end of the evolution experiments so it is often difficult to evaluate how fast the evolution of resistance occurred in these studies. These experiments were conducted during 14 days (ref 55 & 56) that corresponded to approximately 75 to 140 generations (ref 56 & 57), leading to important accumulation of resistance in all cases. In one of these studies (ref 55) the rapid increase of amikacin resistance suggests that some mutations accumulated early in the experiments.

The general trend is therefore that amikacin resistance mutations can rapidly accumulate. It is also the case in our biofilm conditions but again it is difficult to compare our study to these previous studies since the regimens and concentration used are different.

Their planktonic data is very interesting, but I wonder how this aligns with the wider literature?

Answer:

Our results related to planktonic evolution indeed differs from what is currently described in the literature and we have now addressed the results obtained in other studies where intermittent treatment has been used in the introduction section:

line 55: “Adaptive laboratory evolution experiments have been used to reproduce the dynamics of emergence and selection of antibiotic resistance in various bacteria. Exposing planktonic bacteria to sub-inhibitory or progressively increasing concentrations of antibiotics quickly leads to diverse inheritable resistance mutations³. By contrast, the use of periodic and short (3 to 8 hours) lethal antibiotic treatments leads to mutants that have increased tolerance to antibiotics, i.e. their ability to survive but not grow under antibiotic pressure, for instance due to increased lag time or reduction of proton motive force^{4, 5, 6, 7, 8, 9, 10}. In some cases, this increased tolerance favored the emergence of genetic resistance^{11, 12, 13, 14}.”

and in the discussion section

line 392: “ Interestingly, such a relationship between tolerance and resistance, with tolerance promoting the emergence of resistance, has also been demonstrated in planktonic populations treated intermittently with antibiotics ^{11, 12, 14}. However, in our case, tolerance was not provided through selection of *de novo* induced genetic mutations previously reported to increase tolerance in periodically aminoglycoside treated *E. coli* planktonic populations (*i.e.* in *nuoN*, *oppB* or *gadC*) ¹⁰, but by the intrinsic antibiotic tolerance of biofilms. If these tolerance mutations did emerge, the stringency and length of the antibiotic treatment used in our study (24h) could have prevented their selection, by contrast with previous studies using shorter treatment (maximum of 8h) ¹³”

Indeed, in the work of Jan Michiels’s group, planktonic evolution using *E. coli* exposed to intermittent treatment of aminoglycosides resulted in the very rapid selection of tolerant strains. However, in these experiments, the antibiotic concentration is usually lower than the one we used and, probably more importantly, the exposure time is always restricted to 3 to 8 hours while we exposed populations for 24 hours. With such a 24h treatment timing, it may be more difficult to select for tolerant strains, which die slower than a sensitive strain but still are unable to grow in presence of antibiotic.

The claim that amikacin resistance occurred faster in biofilm than in planktonic conditions is troublesome in my opinion as different antibiotic concentrations were used and no data on how much antibiotic actually reaches the bacterial cells in the biofilm exists (esp. what about the bacterial cells that are deeply attached / metabolically inactive etc.).

Answer:

We think that there might be a misunderstanding: we did not use different concentrations of antibiotic between biofilm and planktonic experiments.

We indeed used the same two antibiotic concentrations, either 5x or 80x amikacin MICs (80µg/mL and 1280 µg/mL), to treat both biofilm and planktonic populations. Consequently, the speed of antibiotic resistance evolution can be directly compared between the two lifestyles at 5xMIC and also at 80xMIC.

It should be noted that the different studies that actually compared planktonic versus biofilm evolution towards antibiotics also treated both populations with antibiotic concentration corresponding to the MIC of their ancestral strain (see Ahmed MN et al. 2018, Santos-Lopez A et al. 2019, Trampari E, et al. 2021, Scribner MR et al. 2020).

We have now made this important precision in the revised version of the manuscript, clarifying that the same concentration of amikacin was used to treat biofilms and planktonic populations,

line 137:

“We exposed planktonic cultures or biofilms formed on medical-grade silicone coupons to the same concentrations of 5xMIC (80 µg/mL) or 80xMIC (1280 µg/mL) amikacin during ten cycles of 24 h intermittent treatments (Fig. 1a and 1b).”.

We also acknowledge now in the manuscript that reduction of the amikacin penetration could be one of the mechanisms explaining why biofilms display enhanced tolerance towards amikacin compared to planktonic population. We have actually performed a specific experiment to show that in *E. coli* biofilms amikacin penetration is reduced (see novel Supplementary Fig. S12). This is now indicated

Line 386: "This observed intrinsic tolerance of biofilms could also be partly explained by the possible reduced penetration of amikacin into biofilms as has been shown for *P. aeruginosa* biofilms⁹² and as we showed for *E. coli* LF82 colony biofilms (supplementary Fig. S12)."

Also, important information regarding the potency of amikacin against biofilms is missing - What are the minimum biofilm inhibitory and eradication concentrations? What is the CFU count in a biofilm vs the planktonic cells? Please provide a justification for the use of 5x and 80x MIC values for the experimental evolution experiments.

Answer:

The reviewer has a good point.

We have now measured the MBC for planktonic bacteria (20 µg/mL) and we also repeated the experiment corresponding to Supplementary Fig. S1 using additional concentration of amikacin to measure the MBIC (90% reduction) and MBEC (99.9% reduction) for biofilms (see new Fig S1) and found them to correspond to 48 µg/mL (corresponding to 3xMIC) and 160 µg/mL (corresponding to 10xMIC), respectively.

This information is now indicated in the manuscript in the supplementary methods,

"Minimum Biofilm Inhibition Concentration (MBIC) and Minimum Biofilm Eradication Concentration (MBEC) were defined as the antibiotic concentration that kill 90 and 99.9 % of biofilm cells, respectively, and corresponded to 48 µg/mL (corresponding to 3xMIC) and 160 µg/mL (corresponding to 10xMIC)."

The rationale to choose amikacin concentration corresponding to 5xMIC and 80xMIC was that both concentrations are above the MPC and that at these concentration biofilms actually displayed enhanced survival compared to planktonic population (see Fig. S1).

This is indicated line 137:

"We exposed planktonic cultures or biofilms formed on medical-grade silicone coupons to the same concentrations of 5xMIC (80 µg/mL) or 80xMIC (1280 µg/mL) amikacin during ten cycles of 24 h intermittent treatments (Fig. 1a and 1b). Both treatments correspond to concentrations that are lethal and superior to the mutation prevention concentration (MPC: 64 µg/mL). We showed that biofilms treated with such periodic amikacin treatments displayed enhanced survival compared to planktonic populations (supplementary Fig. S1)."

The 80xMIC concentration was particularly important to use since, in clinical practice, the treatment of biofilm related infections often necessitates the use of high antibiotic concentration.

Figure 2 and supplementary figures: Is the y-axis on this plot log normalized and also normalized relative to survival CFUs of no treatment? The overall representation of 'percentage survival' is confusing and difficult to understand as to what the exact amount of difference between planktonic and biofilm survival is – why not show the real CFU values that were obtained?

Answer:

In the revised version of the manuscript, we now provide more details on the calculation of the survival rate that is measured and represented in the legends of Fig. 2 and other corresponding supplementary Figures.

Survival as well as resistance frequency are plotted on a log-scale to facilitate reading. For each cycle, the survival was calculated by dividing CFU count at step 5 (after treatment) by CFU count at step 2 (before treatment). This was then represented as percentage = $(CFU_{step5}/CFU_{step2}) \times 100$.

No normalization with the survival in the absence of treatment is necessary since the grey data correspond to control where no treatment was given at step 3.

We agree that the CFU counts over time are important data to provide and we indeed provided them in our original manuscript in supplementary figure S3.

We decided not to integrate these CFU counts in the main text, which was already dense, and also because these data do not allow us to easily appreciate the differences between the two life-styles. We consider that survival measurements based on CFU counts are a better representation of the capacity of each population to overcome the treatment as it allows direct comparison between biofilms vs planktonic evolution, which is the purpose of the study. These information have been maintained in the supplementary data.

The methodology how bacterial cells were washed seem problematic (Ln582) – the high centrifugation speed has the potential to damage the bacterial cells which could alter the CFU counts.

Answer:

The washing steps were necessary to remove traces of antibiotic, notably when using high concentration (80xMIC) AMK treatment. Both biofilm and planktonic cells were washed the same way.

This is now more precisely indicated:

line 643:“ Then, bacterial pellets were harvested by centrifuging at 15,000 x g for 3 min at 25 °C, washed once with 200 µL of PBS, and suspended in 200 µL of LB medium to prepare bacterial suspension.»

This washing protocol was performed to allow maximum recovery of cells, which number can be low, particularly after 80xMIC treatment. There was a mistake in indicating centrifugation speed since the provided values are in xg not in rpm. However, the reviewer has a point regarding damaging of cells by centrifugation.

We therefore performed an experiment to evaluate the impact of various centrifugation protocols on the CFU counts (see the graph pasted below). Planktonic stationary phase cells were treated or not with 80xMIC for 24h after which 3 protocols of washing were evaluated: 15000 xg for 3 min (the one used in the manuscript), 10000 xg for 5 min and 6000 xg for 10 min. For each condition, we performed triplicate. The results below clearly indicate that the three methods generated the same level of survival.

The impact of different centrifugation methods on bacterial number in AMK treated samples

What do the authors speculate why there were no mutations in their planktonic cultures considering that A) their culture model let bacteria grow into late stationary phase (nutrient starvation) and B) using glass tubes with potential lower oxygen exposure (oxygen stress)? With the experimental setup alone, one would expect more mutations.

Answer:

As shown in Figure 3, the planktonic population actually also accumulated mutations, including in the non-treated controls. The number of mutations that were strongly selected and reached frequency above the 5% detection threshold of the Breseq analysis was however limited, by the way for both planktonic and biofilm life style.

As far as the use of tubes for planktonic evolution is concerned, we ensured that the tube volume was much larger than the volume of the medium (14 mL tube for 3 mL of medium). Tubes were agitated at 180 rpm and tilted on the rack to allow sufficient agitation to avoid low oxygen stress. Regarding the stationary phase, mutations probably occurred at low frequency, but most of them were probably not selected during the regrowth process that was performed after dilution. When looking in studies that used evolution protocols close to ours with bacteria in some late stationary phase, the number of mutations isolated also remained very low (see, for example, Van den Bergh, B. *et al. Nature Microbiology* 1, 16020 (2016). or Fridman, O. *et al Nature* 513, 418–421 (2014)). Growth advantage in stationary phase (GASP) mutations that appear during prolonged late in stationary phase are selected only after several days of growth (10 days in the 1990's experiments of Roberto Kolter). This is much more than the time they are left in stationary phase in studies like ours or others performing evolution experiments general.

Ln220-222: were any of those mutations confirmed with individual created knockout mutants? How many mutations are overall on the chromosome in these populations? It appears very descriptive without confirmation studies. How did the authors rule out that nothing else caused increased resistance in biofilms?

Answer:

We have focused the present work on mutations that were the most selected during our evolution experiments, *ie sbmA, fusA or fimH*. As indicated in our manuscript *sbmA* and *fusA* mutations have been identified systematically in studies evaluating *E. coli* evolution under aminoglycosides including amikacin. Consequently, there is little doubt that these mutations indeed confer increased resistance to amikacin. We previously identified mutations in the corresponding lectin

domain of FimH as being involved in enhanced biofilm formation (see Yoshida, M. et al. *microLife* **3**, (2022)). One FimH mutation (Δ 29-40) was identical to one mutation we previously characterized in this previous work and the different mutations we identified in the present work were all located in FimH lectin domain as we showed in Supp Fig. S11. Except for some stop codon mutations in SbmA (confirming that this is the inactivation of SbmA that confers increased resistance) none of the mutations in *fusA* or *fimH* were equivalent to knock out mutations, so constructing knock out mutations would have been inappropriate.

However, the reviewer has a point in indicating that direct correlation between one phenotype and a mutation in one gene can only be made if the tested clones have unique mutations. In fact, we actually did correlate directly the observed phenotypes with single point mutations since several of the presented experiments (MIC in Figure 5 and Table S4, growth curves in Figure S9, competitions in presence and absence of amikacin in Figure 6 and biofilm forming capacity/survival in Figure 7) were performed with clones that were fully sequenced and only contained a single mutation in *fusA*, *sbmA* or *fimH* compared to the ancestral strain (see Supplementary Table S4: clones 853, 863 and 891 with single *fusA* mutations G604V, G676C and P610L, respectively; clones 236 and 501 with single *sbmA* mutations Q192L and coding (17/1221 nt) (T)6→7, respectively; clone 239 with single *fimH* Δ 29-40).

The above-mentioned experiments therefore confirmed the link between these single mutations and the observed phenotypes, notably the increased MIC observed for mutations in *fusA* or in *sbmA*.

These aspects were better explained in the legends of the different figures and in line 243-247.

Related to the number of mutations per chromosome: As summarized in Table S2, the end point populations had between 2 and 9 different mutations with frequency higher than 5%. This is now indicated line 193: “End point populations displayed up to 9 different mutations with frequency higher than 5%, relative to the ancestral population».

In addition, while most of the characterized clones had increased MICs, we did not rule out that their presence within biofilms modulated their capacity to sustain the high antibiotic concentration used. The increased MICs of these clones is only a part of the mechanisms that explain their recalcitrance as we are now indicating line 383:

“Biofilm intrinsic tolerance to lethal antibiotic concentration as high as 80xMIC also increases the size of surviving bacterial population within biofilms at each cycle of treatment, while planktonic populations were eliminated after three cycles of such treatment. This observed intrinsic tolerance of biofilms could also be partly explained by the possible reduced penetration of amikacin into biofilms as has been shown for *P. aeruginosa* biofilms⁹² and as we showed for *E. coli* LF82 colony biofilms (supplementary Fig. S12).”

Furthermore, we discussed the fact that the fitness cost of these resistance mutations could be reduced in the biofilm conditions we used compared to the one in planktonic conditions: See paragraph starting from line 561.

Regarding the early selection of sbmA and fusA mutations in biofilm – how relevant would this be in an established (thick) biofilm in a patient?

Answer:

Because biofilms still remained relatively difficult to detect in clinical situation, the relationships between the size (or maturity) of in-patient biofilms and the necessity to treat the infection is not clearly established. It is therefore difficult to determine the exact relevance of the “early” selection of resistance mutations in biofilms.

However, as stated above, there is an emerging recognition that biofilms responsible for infection might not actually correspond to very mature biofilms, yet symptoms might still be apparent in these situations because of local inflammation and shedding of bacteria in surrounding tissues or in the circulatory systems, and thus administration of antibiotic treatment could lead to situation of selection of resistance mutations. Moreover, even if in-patient biofilms become more mature, we can expect that this could increase the diversity of resistant mutations due to biofilm high population size and mutation frequency and because of the enhanced tolerance associated with biofilm formation. In addition, as we showed for *fusA* mutations, even resistant clones with a fitness cost that would have been lost in a planktonic lifestyle could be maintained in a biofilm. Consequently, formation of more biofilm biomass would constitute a proportionally more important reservoir of resistance mutations to be selected for upon antibiotic treatment.

Other minor comments:

Ln60: experiments

Answer:

This has been corrected.

Ln63-64 / Ln117: Mutations that lead to increased tolerance need to be further explained as it is not clear what the authors refer to. If cells 'just' survive, how can this lead to biofilms have a 'high mutation rate' as stated in the abstract?

Answer:

The previously identified tolerance mutations that we are referring to are genetic mutations increasing the survival of the strains without changing their MIC as defined in Brauner et al *Nat Rev Micro* **14**, 320–330 (2016). There are multiple different mutations that were identified to cause enhanced tolerance in the cited studies. Because we cannot cite them all, we just gave two examples of the consequence of such mutations.

See line 58: "By contrast, the use of periodic and short (3 to 8 hours) lethal antibiotic treatments leads to mutants that have increased tolerance to antibiotics, *i.e.* their ability to survive but not grow under antibiotic pressure, for instance due to increased lag time or reduction of proton motive force^{4, 5, 6, 7, 8, 9, 10}. In some cases this increased tolerance favored the emergence of genetic resistance^{11, 12, 13, 14}".

Tolerance mutations and high mutation rate are not two antinomic phenomena especially in biofilms. Heterogeneity is one of the characteristics of biofilm physiology. Consequently, in biofilms certain bacteria can be tolerant because of their protection within deep layers of the biofilm or by their reduced growth or because they accumulate tolerance mutations while others can be directly subjected to a variety of different stresses, which in turn increases the mutation rate. Hence, the high genetic diversity found in biofilms constitute a reservoir of mutations potentially selected upon exposure to a new stress. Interestingly, there is even a link that has been established by the Michiels's team between persistence (represented by a subpopulation of tolerant cells) and enhanced mutation rate (Windels, E. M. *et al.* doi:10.1038/s41396-019-0344-9).

We have now added some text to better reflect the multiplicity of mechanisms explaining enhanced biofilm survival and selection of resistance in the introduction section:

line 119: "Our study therefore shows that, when biofilms are exposed to clinically relevant intermittent lethal antibiotic treatments^{51, 52}, enhanced biofilm tolerance combined with high mutation rates leads to the rapid selection of clones with increased MIC to amikacin. This suggests that biofilms provide a heterogenous environment where higher mutation rates and intrinsic tolerance promote the survival of otherwise counter-selected resistance mutations, potentially contributing to the antibiotic crisis." ; and line 441 at the end of the discussion

“Altogether, our study demonstrates that biofilms exposed to lethal intermittent treatment rapidly evolved towards enhanced survival to antibiotics through multiple mechanisms involving intrinsic and promoted tolerance and/or enhanced mutation rates that may favor the rapid selection and maintenance of high cost antibiotic resistance mutations”.

Ln65: “before to favor” = “before favoring”

Answer:

This has been corrected.

Ln65: remove “genetic”

Answer:

We prefer to keep “genetic resistance” since the concept of resistance (without the added term genetic) could be vague and misinterpreted and we want to refer to true genetic resistance.

Ln66: add ‘a’ before clinical setting”

Answer:

This has been changed to “Worryingly, this observation was confirmed with patients treated ...”.

Ln68: plural - infections

Answer:

We kept the singular since one can consider that the same infection was maintained along the patient history and not multiple infections.

Ln71-74, 79-80: Based on that thought, traditionally, those bacterial species isolated from biofilm-associated infections were antibiotic susceptible in the lab – so how does this contribute to the overall ‘selling point’ of this manuscript?

Answer:

The purpose of these two assertions is to emphasize the importance of studying biofilms because of their medical importance notably through their link with infection chronicity (line 67-72) and to provide some information on the possible mechanistic beyond the observed increased mutation rate in biofilms (line 75-78). We consider that this information is important to contextualize the study. Because biofilms display already a high tolerance that can explained chronicity of infection the potential emergence of true genetic resistance could constitute an aggravating factor that may complicate treatment and lead to more severe chronicity.

An additional sentence was added line 70 to take this comment into account: “The emergence of antibiotic resistance within a tolerant biofilm population could therefore constitute an aggravating factor increasing the frequency of therapeutic failure and infection recurrence.”

Ln81-83: this should be further explained with clear examples

Answer:

It is difficult to provide more information for the cited *E. coli* studies since, whereas the authors clearly demonstrated enhanced MICs to different antibiotics, they did not sequence the evolved population, so there is no description of the mutations that appeared.

For the *S. aureus* study, we decided to remove this reference since the process of VISA resistance appeared in complex interactions within biofilms, and we feel that more information will rather complexify our message, which is that biofilms have very strong evolvability and can probably accumulate a range of various mutations even in absence of specific selective pressure.

We now just provide additional information on the fact that resistance could emerge against various antibiotics of different classes.

line 78: "Consistently, *Escherichia coli* biofilms rapidly acquire resistance to several different classes of antibiotics, even in the absence of antibiotics^{36, 37, 38}."

Ln90: "biofilm high antibiotic tolerance" change to "biofilm associated high antibiotic tolerance"

Answer:

This has been corrected.

Ln92: "speed" = "rate"

Answer:

This has been corrected.

Ln94: remove "colony"

Answer:

We kept the term "colony biofilms" since this referred to a specific model of biofilm formation whereby bacteria are spotted on agar or generally nitrocellulose disk deposited on agar to grow as very large colony. This model is well known in the biofilm community.

Ln104-106: This sentence needs to be rephrased to mention the comparison of biofilm to planktonic growth as there is no doubt that biofilms "constitute a dynamic evolutionary reservoir enabling the emergence of high antibiotic resistance"

Answer:

This has been corrected according to your suggestion.

see line 100:

"Hence, it is still unclear whether, compared to planktonic bacteria, antibiotic-tolerant biofilms constitute a more dynamic evolutionary reservoir enabling the emergence of antibiotic resistance."

Ln75-106: overall this paragraph sound more like a discussion than introduction

Answer:

We do not agree with the reviewer. The purpose of these two paragraphs is to provide a presentation of the previous work performed on the main characteristics of biofilms related to antibiotic treatment and their important evolvability. In these two paragraphs we acknowledged the work of other teams showing that biofilms could evolve resistance in different conditions but that when compared to the evolution of antibiotic resistance biofilms were not always evolving faster or stronger resistance. Then, this latter statement allows us to indicate that it is still unclear

whether or not antibiotic-tolerant biofilms could constitute a more dynamic evolutionary reservoir enabling the emergence of high antibiotic resistance than the planktonic lifestyle, thus implying that more comparative studies are necessary.

Therefore, we would rather keep this presentation of the current state of the field in the introduction section.

Ln123: can the authors reference the clinical relevance accordingly?

Answer:

Corresponding references were added.

Ln143: remove "as"

Answer:

This has been corrected.

Ln144-148: What does this mean and what are the authors trying to communicate?

Answer:

Because the time scale of evolution is highly related to the number of generations during which mutations occur, it is mandatory to provide these numbers of generation to be able to compare the evolution rate between planktonic and biofilm population. Our evolution experiments correspond to less generations in biofilms than in planktonic population, and thus biofilms evolved less longer in term of generation than planktonic cells. Therefore, it is possible to conclude that the apparent fastest selection of resistant mutants in biofilms is not due to a longer evolution time compared to planktonic population.

To improve readability, this information has been moved to the Material and Methods section and There is a comment related to this aspect:

line 182 "Despite being propagated with a lower number of generations (supplementary Table S1), biofilms therefore evolve resistance faster and at higher frequency than planktonic populations."

Fig 2(C/D): I am unsure if showing the MIC mean and standard errors of the mean for 3 independently evolved lines is appropriate. How many times was the MIC determined for each evolved line at each step? Was there variability within this number? Additionally treating these independently evolved lines as if they are a replicate set is slightly misleading, perhaps showing each independent line MIC value at each cycle would be more appropriate.

Answer:

The MIC of each evolved lineage was determined in triplicate (n= 3) and there was no variability of these MIC between these three replicates.

This is now indicated in the Material and Methods section and in the corresponding figure legends.

We chose to represent an average of the MIC for the three independent evolution lineages of each lifestyle to facilitate figure reading. However, as indicated in the text for each of the panel of Fig. 2 we are providing the CFU, MIC and frequency of clones growing on plates plus amikacin at each cycle for each independent lineage in Supplementary Fig. S2. The variability between each evolved lineage can be then visualized in this supplementary figure.

Fig 3, Fig S6, Table S2: What was the rationale of selected genes (also gene names must be italicized) and how many samples were used for frequency calculations?

Answer:

The rationale of the presented genes is based on the fact that the corresponding population at the end of the experiment contains mutations in a specific gene with a frequency higher or equal to 5%. These frequencies of mutation were calculated using the Breseq program performed on the Illumina reads originating from each end-point population. The sequencing of each population was performed once, the depth of sequencing (average of x150) being enough to ensure the high quality of the results.

We have added some information to specify some of the missing aspects in the corresponding figure legends and in the material and methods section.

Gene names were italicized in both figures (see below to see how the figures look like after this modification for Fig. 2).

Moreover, although it is clearly mentioned that planktonic lines 4-6 (exposed to 80x MIC) did not survive the entire 10 selections, it is stated that those populations were sequenced at their last survival stage. Where is the data for this? In Table S2 it states they were not determined, is this correct, or was there not detectable mutations in the populations above 5% frequency?

Answer:

The reviewer is right, the populations were sequenced at their last survival stage (cycle 2 for population 4 and 5, and cycle 3 for population 6). There was no detectable mutation in the populations 4 and 6, but one fixed mutation in population 5.

This information is now available in revised supplementary Table S2 and S3.

Regarding the colouring of these figures: It is very difficult to distinguish mutations which occur rarely from those which do not occur at all (eg: yhdP in control line 9, in figure 3). I would suggest going from one colour to another (not white to red), or perhaps more appropriately putting a % value on each square with a value above 5% so the reader can clearly distinguish boxes which have for example 20% mutation frequency and those which have 50% (much like is done for Figure 5).

Answer:

Both figures were changed accordingly. Thank you for suggesting this improvement. See for example the modified Figure 2:

Ln168-169: Using the word “evolution” when talking about increases in MIC confuses the terminology, as the MIC itself is not evolving it is symptom of the bacteria evolving resistance, I would stick with just using terms like increase.

Answer:

We agree with the reviewer. The description of MIC was corrected throughout the manuscript with MIC modification or increase, instead of MIC evolution.

Ln172-177: The use of the word “clones” here is inconsistent with the terminology being used in the figures either rephrase to call them “resistant clones” or “resistant mutants”.

Answer:

This has been corrected according to your suggestion.

Ln188-189: Although an accurate statement in the context of the whole manuscript -It is not until you sequence individual clones (figure 5) that you could have been certain that the survival you were observing was caused by resistance and not a combination of resistance and tolerance, perhaps this sentence is premature in the manuscript.

Answer:

Thank you for this comment.

This statement is based on the fact that the MIC of biofilm populations is rising faster and at higher level than planktonic ones, and so is the resistance frequency. Therefore, resistance is clearly increasing faster and at a higher level. However, the reviewer is correct to point that this does not mean that resistance is the only parameter involved in the underlying evolutionary dynamic, and we did not intend to mean that.

Therefore, we have revised our text as below:

Line 182

“Despite being propagated with a lower number of generations (supplementary Table S1), biofilms therefore evolve resistance faster and at higher frequency than planktonic populations.” To us this statement does not exclude that tolerance could also be at play in the observed increased survival.

Ln208-211: FimH mutations are also seen in planktonic cultures without antibiotics

Answer:

The sentence has been modified as follows

Line 206 “Finally, almost all evolved biofilm populations displayed mutations in the gene coding for the FimH tip-adhesin of type 1 fimbriae at a higher frequency than in planktonic populations⁶⁰”

Figure 4: gene names must be italicized

Answer:

In this figure, the legend shows the amino-acid modifications at the protein level.

Ln410: Incorrect supplementary figure referred to – should be S12.

Answer:

This has been corrected.

Ln416-429: this discussion is mainly speculation and does not add much; could be deleted

Answer:

Thank you for drawing our attention to this point. We agree with you that we are speculating here. However, since we find that presenting these hypotheses at this point of the discussion is nevertheless relevant, we removed them from the main discussion but kept it as supplementary Discussion in the Supplementary Material section.

Fig 8: the increased biofilm formation is questionable based on those low numbers and large overlap (the CFU counts are difficult to understand as the y-axis scale is not appropriate for this data).

Answer:

The reviewer's comment seems to rather correspond to Fig. 7.

The reviewer has raised an important point; however, we believe that biofilm formation OD600 and CFU numbers in Fig. 7A and B, although relatively low, are reliable, with differences supported by statistical analyses revealing significant difference between FimH mutants and the corresponding wt ancestral isogenic strains. The observed enhanced biofilm formation caused by mutations in *fimH* when evolving biofilm is also coherent with what we previously demonstrated (see Yoshida M et al, *microLife* 3, uqac001 (2022)):

and stated line 310

"We recently showed in a previous study that mutations enhancing adhesion to abiotic surfaces are rapidly selected in the type 1 fimbriae tip lectin FimH during biofilm formation, including in frame deletions⁵⁰. Considering the increased intrinsic tolerance of biofilm to antimicrobials, we hypothesized that the high frequency of mutations identified in *fimH* in our evolved biofilm populations—all located in the FimH lectin domain (supplementary Fig. S11)—could also contribute to increase biofilm tolerance and survival (Fig. 3)."

This enhanced biofilm formation is also consistent with the observed survival to a lethal high concentration of amikacin. Hence, we would like to retain the Fig. 7 as it is.

Concerning the survival rate provided in Fig. 7C, we still consider that this is much more meaningful than providing solely the CFUs. As indicated in the Material and Methods section and now in the legend of Fig. 7, they have been calculated by using the formula $100 \times (\text{CFU}_{\text{AFTER}} / \text{CFU}_{\text{BEFORE}})$, where $\text{CFU}_{\text{BEFORE}}$ corresponds to CFU/coupon before treatment and $\text{CFU}_{\text{AFTER}}$ to CFU/coupon after treatment.

*Ln431: It should be noted also that mutations in *fusA1* in *P. aeruginosa* have been shown to confer resistance to aminoglycosides (including amikacin) alongside being commonly selected for during infection (see Bolard et al, 2018. 10.1128/AAC.01835-17).*

Answer:

We have added the reference according to the reviewer's suggestion.

*Ln475: Are the high rate of *fimH* mutations (leading to enhanced biofilm formation) not evidence of increased antibiotic tolerance mechanisms alongside antibiotic resistance mechanisms contributing to survival and resistance?*

Answer:

Yes indeed, the reviewer is totally right. We are now better acknowledging this point

line 312:

"Considering the increased intrinsic tolerance of biofilm to antimicrobials, we hypothesized that the high frequency of mutations identified in *fimH* in our evolved biofilm populations—all located

in the FimH lectin domain (supplementary Fig. S11)— could also contribute to increase biofilm tolerance and survival (Fig. 3)”

and line 318: “

“Therefore, the differential accumulation of *fimH* mutations in evolved biofilm populations also likely contributed to enhanced tolerance to otherwise lethal antibiotic concentrations.”, as well as in the discussion section line 436 at the end of the paragraph related to FimH: “The selection of such biofilm-promoting mutations could therefore be seen as contributing to a positive feedback loop, enhancing biofilm formation and de facto enhancing tolerance and survival of biofilm bacteria, which would increase the odds for de novo genetic resistance mutations to occur.”.

Ln582: please state x g not rpm

Ln590: please state x g not rpm.

Ln723: please state x g not rpm.

Answer:

rpm has been replaced by x g.

Ln702: remove space between °C

Answer:

This was modified.

Ln737: How were sequences trimmed, were they quality and adapter trimmed, using which software?

Answer:

Reads quality was assessed using FastQC version 0.11.9 (<http://www.bioinformatics.babraham.ac.uk/projects/fastqc/>) and trimmed using Trimmomatic version 0.39 (Bolger et al., 2014). This was added to the methods section. We apologize for this oversight.

Ln741: please add version of lollipop used (if one exists)

Answer:

The version was added accordingly:
“... lollipop version 0.9.0...”.

Ln742: please correctly cite ggmuller (see the readme for ggmuller on instructions for this - <https://cran.r-project.org/web/packages/ggmuller/readme/README.html>)

Answer:

This was modified with the reference: Robert Noble (2019). ggmuller: Create Muller Plots of Evolutionary Dynamics. R package version 0.5.3. doi:10.5281/zenodo.591304 <https://CRAN.R-project.org/package=ggmuller>.

Ln789: please state x g not rpm.

Answer:

This does not correspond to centrifugation as this value refers to the rotation speed of the shaking plate in the incubator, which is literally expressed as rotations per minutes and not in g.

Ln856: Thank you for making all code used available, please ensure before publication the BioProject is released.

Answer:

NCBI will release them automatically upon publication.

Ln877: Please ensure consistency across references included, some contain DOIs and other do not.

Answer:

Only one reference included a DOI. This paper was in press at the time of the original submission. It is now published and the full reference is given.

Fig S9: How many times/replicates were done for each growth curve.

Answer:

Results are shown as mean from the measurements of 3 biological replicates for each strain. This is indicated in the corresponding Material and Methods section as well as the legend of the corresponding figure.

Reviewer #2 (Remarks to the Author):

Masaru et al. carried out a tour de force study comparing evolution of amikacin resistance for pathogenic *E. coli* LF82 grown under biofilm or planktonic conditions when treated with intermittent lethal levels of antibiotic. Masaru et al. observed bacteria accumulated more mutations, mainly in *sbmA* and *fimH* genes, when grown in a biofilm compared to planktonic growth conditions. In particular, the authors notice that *fimH* is rarely observed for bacteria when grown under planktonic conditions, which the authors attribute to imparting a fitness cost for bacteria grown under planktonic conditions. Overall, this is an impressive amount of work with high quality data. However, it can be difficult to follow sometimes due to technical details and phrasing, and as a result, may be more appropriate for a specialized journal.

Answer:

Thank you for these positive comments. We have simplified some technical aspects to facilitate reading and the manuscript has been thoroughly corrected by a native English speaker.

General comments:

It is still not clear why bacteria grown in a biofilm accrue more mutations compared to bacteria grown under planktonic conditions. The authors should carry out sequencing studies to confirm this by growing bacteria under the two conditions, but without antibiotics.

Answer:

The fact that biofilms are more prone to accumulate mutations (and diversity) compared to planktonic populations is well recognized in the field, and proposed to be essentially related to stress responses that are encountered by bacteria when growing in biofilms conditions, where they are exposed to nutrient stress, oxygen gradient, etc.

This was indicated in our introduction

line 75: “Physico-chemical heterogeneity and diffusion limitation in biofilm environments may reduce exposure to antibiotics²⁹, induce bacterial stress response and increase mutation rates compared to planktonic conditions, leading to genetic diversification^{30, 31, 32, 33, 34, 35}. Consistently, *Escherichia coli* biofilms rapidly acquire resistance to several different classes of antibiotics, even in the absence of antibiotics^{36, 37, 38}.”

As shown in supplementary Fig. S10 we also demonstrated that the biofilms formed in our study displayed a higher mutation rate.

This is mentioned line 303: “Alternatively, or additionally, *fusA* mutations could have accumulated at higher frequency in the stressful and mutation-prone biofilm environment^{34, 66}. Consistently, we showed that the mutation rate in the biofilms formed in our experimental evolution set-up was higher than in planktonic conditions (supplementary Fig. S10), therefore providing increased opportunity for genetic diversification and the rapid apparition and selection of resistance mutations.”

*The authors place a lot of emphasis of phenotypes observed when a *sbmA* mutations are not present. However, there are other mutations present that could also explain the phenotype.*

Answer:

Our study indeed focuses on resistance genes found mutated at high frequency and/or identified in different biofilm and planktonic populations. The consistent identification of similar mutations (in *sbmA*, *fusA* and *fimH*) is indicative of convergent evolution and shows that these genes are the main drivers of evolution towards resistance to amikacin in our experimental evolution

experiments. The analysis of clones that contained a single mutation in these genes (*sbmA*, *fusA* and *fimH*) compared to the ancestral strain also supported our hypothesis that these mutations were the main drivers of the observed evolved phenotype (see Supplementary Table S4: clones 853, 863 and 891 with single *fusA* mutations G604V, G676C and P610L, respectively; clones 236 and 501 with single *sbmA* mutations Q192L and coding (17/1221 nt) (T)6→7, respectively; clone 239 with single *fimH* Δ29-40).

Additionally, we did not neglect the potential contribution of mutations in other genes since we provided information on other mutated genes and their potential link to enhanced resistance.

See for instance between line 247 and 264 for mutations found in biofilm evolution and a specific paragraph in supplementary discussion entitled “Mutations leading to moderate MIC increase in planktonic clones are diverse” for mutations found in planktonic evolution.

Specific comments:

Line 104: ‘Hence, whether or not antibiotic-tolerant biofilms constitute a dynamic evolutionary reservoir enabling the emergence of high antibiotic resistance is still unclear’

Based on such different results observed in the literature, it seems like evolution of resistance is very context dependence and maybe there isn’t a universal explanation (i.e., biofilm always evolves resistance faster or at greater diversity compared to planktonic, or vice versa).

Answer:

We fully agree with reviewer’s comment. Each study has its own biofilm formation method, which, in addition to the fact that different species are used, probably leads to different biofilm structure. However, in most cases and despite the use of different biofilm methods, different bacteria, different treatment regimens, there is a strong tendency for biofilm, notably through their enhanced tolerance, to favor the selection of antibiotic resistance.

We now acknowledged this aspect at the end of our discussion section:

Line 444 “Whereas this phenomenon might be context dependent, our work contributes to the demonstration that antibiotic tolerant biofilm environments generally promote the worrisome evolution of antibiotic resistance^{36, 39, 40, 41, 42, 43, 44, 95}”

Line 116: please list what are the typical mutations observed.

Answer:

The typical mutations are now listed as follows:

Line 112 “By contrast with what was observed with shorter periodic treatments of planktonic populations, neither biofilm nor planktonic bacteria accumulated mutations previously associated with tolerance to aminoglycoside (i.e. in *nuoN*, *oppB* or *gadC*)¹⁰. Instead, biofilm populations accumulated mutations in type 1 fimbriae tip lectin FimH promoting biofilm formation⁵⁰, which, in turn, promoted biofilm-associated tolerance..”

Line 128: please list proposed alternative treatment options (i.e., different treatment regimes, peptides, phages, exc.).

Answer:

It is difficult at the current stage of our study to clearly identify alternative strategies. The use of novel antimicrobial peptides or of phages are certainly promising emerging strategies but their development does not rely on experiments that demonstrated evolution of resistance under certain treatment regimens. We rather imagine that studies such as ours could be useful to warn

clinicians about the risk of emergence of resistance using certain type of treatments and then might lead to adapting the current treatment regimens or developing novel treatments.

We have amended the end of our introduction accordingly:

Line 124 “Our study may inform the modification of current biofilm infection treatments to reduce the risk of antibiotic resistance emergence and disease recurrence by providing new insights into the evolution of antibiotic resistance in biofilms.”

Line 142: ‘superior to the mutation prevention concentration (MPC: 64 µg/mL)’: how was this concentration determined. You are clearly observing mutations enabling resistance so it is not mutation prevention?

Answer:

The reviewer raised an interesting point. The method we used for measuring MPC is described in Materials and Methods and was determined on planktonic population as per recommendations, and there was therefore no cycling of treatment and recovery.

In our study, evolution experiments were conducted with intermittent exposures at concentrations exceeding the MPC and recovery period, and we essentially observed that biofilms (but also planktonic population at very late stages of the evolution) selected for resistance mutations.

There are several potential explanations.

First, antibiotic diffusion and penetration could be reduced in biofilms (as we showed in response to one question of reviewer 1) it is therefore possible that in some biofilm areas, the actual antibiotic concentration becomes lower than the MPC, creating very favorable conditions for cells to evolve resistance even when concentrations used are above the MPC. It is worth noting that planktonic population could not sustain x80MIC for more than 3 cycles demonstrating clearly the much higher recalcitrance of biofilms.

Second, all concentrations such as the MIC, MBC, MPC, MBIC and MBEC are all measured on precise population sizes and for MIC, MBC, MBIC and MBEC, correspond to between 90 to 99,9% eradication of the population. It is possible to reach higher bacterial concentration in biofilms and in late stationary planktonic cells, and because of these large populations there are always bacteria that will survive, recover and potentially evolve resistance during the regrowth phase between each treatment.

This aspect is now indicated both in the results section:

line 171 “It should also be noted that resistance evolved in both biofilms and planktonic populations despite the use of above MPC antibiotic concentrations.”

And in the discussion section,

line 351 “It should be noted that the evolution of resistance in our experiments, despite the use of antibiotic concentrations above the MPC, can be explained by the large populations present in our settings. This may have led to the absence of full eradication (notably in biofilms) and the capacity of bacteria to resume growth and mutate during the recovery phase between treatment cycles.”

Figure 1: incomplete data for planktonic 80x. The authors mention doing ~200 plus cycles, but only show data for 3 cycles.

Answer:

Thank you for your comments.

~200 corresponds to generations, not cycles. Planktonic population did not survive more than 3 cycles of treatment with 80x MIC, by the way reflecting their much higher sensitivity compared to biofilms. Therefore, there was no available data for these planktonic lineages after the third cycle. Since the decrease of planktonic population was important at each of these 80x MIC treatment and the regrowth allowed for 48 h (see Supplementary Fig. S3B) then these lineages evolved for ~212 generations during these 3 cycles (see information now in Materials and Methods section).

Figure S3 should be moved into the main text. It is very informative having the raw CFU counts instead of %. It rules out the possibility that there are simply more bacteria for biofilm conditions, and that is why more mutations are observed.

Answer:

We fully agree with the reviewer that the raw CFU counts are very important to provide (and this is why we gathered them in Supplementary Fig. S3). However, to avoid increasing the length of our manuscript and because we consider that providing the % of survival are much more informative to carry out biofilm vs planktonic comparative study, we prefer to maintain the raw CFU data as supplementary information.

Figure 3: Why are the replicates labeled 1-9? Labeling it as replicate 1-3, 1-3, 1-3 would be clearer. Also, frequency should be on the scale of 0-1.

Answer:

To clarify the labeling of the different lineages we have modified the names of these lineages for B1 to B9 for biofilms and P1 to P9 for planktonic in all the corresponding tables and figures.

Concerning the frequency of the mutations in the populations, these frequencies correspond to the direct output of the Breseq program that is detecting mutations that are at a percentage (or frequency) higher than 5%. Following a request from reviewer 1, we have modified this figure and the corresponding figure S6 by adding the corresponding frequency in each box.

Line 210: 'Although these mutations are not directly associated to antibiotic resistance, they might increase biofilm formation and have an impact on survival against antibiotics (see below).' Please continue the discussion, rather than having to fish for it later. Or add, see below in section.'

Answer:

We prefer keeping a specific section to present results related to FimH. We thus indicated "see below in section related to FimH".

Line 216: It isn't clear what 'sample 3' is referring to (i.e., Figure 3 or Figure 4). If Figure 3, could the explanation be the presence of another mutation (fadE) rather than the absence of a sbmA mutation not leading to enhanced resistance? If referring to Figure 4, there is no sample 3 for planktonic condition.

Answer:

We have modified this part of the results to clarify it:

See line 214: "Moreover, we confirmed the link between presence of *sbmA* mutations and enhanced population resistance: planktonic populations P1 and P2 contained mutations in *sbmA* and displayed an increased MIC towards amikacin, whereas planktonic population P3 had no

mutation in *sbmA* and no such increase (Fig. 3 and 4). Population P3 also had the lowest number of colonies growing on amikacin containing plates (supplementary Fig. S2).”

Line 230: How was this determined, by increased MIC or literature precedent: ‘enhanced genetic antibiotic resistance – and not tolerance –’ ? If by MIC, please include rationale in sentence.

Answer:

This sentence is the conclusion of the corresponding results section and a precedent described in the literature precedent. We have changed this sentence for:

Line 228 “Our analyses therefore showed that intermittent treatment with lethal concentrations of amikacin led to enhanced antibiotic resistance reflected by an increase in amikacin MIC and frequency of resistant clones. This demonstrated that the early selection of mutations in *sbmA* and *fusA* are the main drivers of the observed enhanced biofilm survival. By contrast, the weak enhanced resistance of evolved planktonic population emerging in late treatment cycles correlates with rare *sbmA* mutations at a relative lower frequency.”.

Reviewer #3 (Remarks to the Author):

Usui et. al present a study investigating the evolutionary trajectory of amikacin resistance in planktonic and biofilm cultures. Evidently, the study of resistance evolution is of great clinical concern and the insights gained may benefit improved antibiotic stewardship. Using parallel sequential passaging with intermittent antibiotic exposure, the group finds that the biofilm state selects for more rapid and potent resistance mutations than in planktonic cultures, suggesting that the nature of infection can impact the emergence of resistant clones. Altogether, the manuscript presents a compelling investigation of bacterial evolution of antibiotic resistance.

Answer:

Thank you for these positive comments.

However, the work could benefit from additional experiments that establish a stronger causal link between specific mutations and fitness/resistance outcomes. These additional experiments would add more biological insight into the findings. Nonetheless, the observation of differential evolutionary trajectories of resistance mutations in different cellular lifestyles is compelling.

Answer:

We have simplified some technical aspects to facilitate reading and the manuscript has been corrected by a native English speaker.

General Comments

- The quality of writing is generally quite good, but there are typos and misuse of word choice throughout that detract from the understandability of the article. I would recommend further editing, particularly in the introduction.

Answer:

Based on these comments, we tried to more carefully edit the revised version of the manuscript, which has also been reviewed by a native English speaker. We also have streamlined some aspect of the text to improve readability.

o Ex: Line 38: "much less is known about the development [...]"; Line 41-42: "relevants cycles of lethal treatment"; Line 59-60: 'The use of adaptive laboratory experiments'. Etc

Answer:

Apologies, those are indeed grammatical errors we should have pick up. The revised version was reviewed throughout the manuscript and further proofread by a native English speaker.

- The hypothesis that the biofilm state selects for different and more potent amikacin mutations as planktonic growth is supported by the evolution experiments. However, I believe these observations would be significantly strengthened if a third sequential passaging experiment was performed wherein the cultures were made to oscillate between the coupon-biofilm growth and planktonic growth. In that case, one could observe if the cost of the resistance mutations in planktonic growth outcompetes the benefits of resistance that arises during biofilm growth.

Answer:

We agree with the reviewer suggestion and we have actually already started to perform such sequential oscillating experiments to study the interplay of selection forces between adhesion and tolerance versus antibiotic resistance.

We hope, however, that the reviewer can recognize that this is a project on itself, representing a large body of work, almost equivalent of what is already included in the current manuscript. Hence, although this will certainly be interesting, we believe this new amount of data would be difficult to intelligibly merge into the current study and should be considered as out of the scope of our current submitted study.

In addition, we would also like to argue that our biofilm evolution experiments were designed and conducted to expose biofilm (and planktonic) bacteria to antibiotic regimens mimicking clinical biofilm infection situations. The study of the suggested oscillation phenomena is unlikely to exist in clinical situation and therefore would not fit well in the submitted study.

*- A major concern of the paper are the biological conclusions drawn from the whole genome sequencing of individual clones from the passaging experiments. While it's clear that the *sbmA*, *fusA*, and *fimH* mutation likely impact amikacin resistance, it is difficult to demonstrate causality without comparison of these mutations in otherwise isogenic strains. Based on the results in Figure 7, it seems like the group has the capacity to genetically modify the *E. coli* strains and perform the MIC/competition experiments with interesting *fusA* and *sbmA* mutations. With these experiments, much stronger and more informative claims can be made on the sufficiency of these mutations to confer increased resistance or biofilm tolerance or if these mutations are only relevant in the context of additional mutations in the genome.*

Answer:

We agree with reviewer 3 that causality can be fully demonstrated by comparing single mutations in otherwise isogenic strains.

We did perform the proposed experiments as clones were fully sequenced and showed to only differ from the ancestor strain by only a single mutation and were therefore otherwise isogenic.

While this was not probably clear enough in our original version of the manuscript, please see MIC measurement in Figure 5 and Table S4, growth curves supplementary Figure S9, competitions in planktonic and biofilm lifestyle in presence and absence of amikacin in Figure 6 and biofilm forming capacity/survival of *fimH* mutants in Figure 7. These experiments were performed with clones 853, 863 and 891 carrying single *fusA* mutations G604V, G676C and P610L, respectively; clones 236 and 501 carrying single *sbmA* mutations Q192L and coding (17/1221 nt) (T)₆→7, respectively; clone 239 with carrying single *fimH* Δ29-40 mutation (see Supplementary Table S4 for details of the mutations detected in each clone).

The provided experiments therefore confirmed the link between these single mutations and the observed phenotypes, notably the increased MIC observed for mutations in *fusA* or in *sbmA*. These aspects were better explained in the legends of the different figures and in line 243-247.

Moreover, of course, direct causality between *sbmA* and *fusA* mutations and enhanced resistance to aminoglycosides has been established in the literature and cited in our manuscript, as well as the causality between FimH mutations and enhanced biofilm capacity (Yoshida et al. 2021).

Specific Comments

*- Line 198-199: *SbmA* is described as an inner membrane transporter associated with increased resistance to aminoglycosides. One would expect loss-of-function mutations to sensitize strains*

to amikacin, but it seems like the opposite is observed in the experiments. Please add some additional discussion to better clarify the purported nature of these mutations.

Answer:

Thank you to drawing our attention on this aspect. We indeed need to clarify that previous literature associated loss-of-function mutations of *sbmA* with increased resistance and not the presence of SbmA itself as a protein favoring resistance.

We actually provided some examples in our study confirming this aspect. For example, the *sbmA* mutation 17/1221 nt) (T)6→7 present in clone 501 will modify the codon reading frame of the protein thus generated a defective protein and MIC of 501 clone is enhanced. Along the evolution in biofilms we also selected stop codon mutants of *sbmA* (for example W111* and W99*, see population B3, B4, B5 and B6 in Fig.4), which is also an indication that these are *sbmA* loss-of-function mutations that are selected.

The association of loss of function of *sbmA* mutations with increased resistance is now clarified as follows:

line 194: "We identified mutations in *sbmA* in all treated biofilm populations and 2 out of 3 treated planktonic populations, but not in control ones. SbmA is an inner-membrane peptide transporter, which loss of function was previously associated to increased *E. coli* resistance to aminoglycosides, including amikacin^{55, 56, 57}. Most populations with *sbmA* mutations displayed several *sbmA* mutated alleles, including stop codon mutations, at various frequencies, indicating potential clonal interference between these different alleles (supplementary Fig. S6 and supplementary Table S2). This was particularly evident in biofilm evolved populations, where 5 out of 6 evolved populations displayed 2 to 5 *sbmA* mutated alleles, regardless of treatment concentration".

- The discussion of the contents of Fig. 7 in the Results are inadequate. Are these naturally occurring mutants from the evolution experiments? Or have they been generated through genetic modification?

Answer:

We acknowledge that this part could appear complex due to the genetic approaches that we had to use to produce one of the isogenic mutants. In the experiments presented in Figure 7, we used a fully sequenced clone (clone 239) that has a single mutation in *fimH* (Δ 29-40) and is otherwise isogenic with the ancestral strain. For *fimH* mutations L79R that also appeared during our evolution experiments, we did not find any clone that just only contained this mutation but we had a clone (219) that had both non-synonymous *fusA* G604V and *fimH* L79R mutations. We therefore deconstructed (repaired) the *fusA* mutation in this 219 clone by allelic exchange into clone 219 to restore the wt copy of *fusA* in this clone. So L79R and Δ 29-40 *fimH* mutants can be directly compared to the ancestral strain

We have tried to clarify this in the Material and Methods section and in the legend of the corresponding Fig. 7.

Is the truncated mutant expected to phenocopy the point mutation? Please expand the discussion of results.

Answer:

Concerning the phenocopy between the Δ 29-40 deletion and the L79R FimH mutations. Yes, we expected both mutations to phenocopy each other because they have both been selected during

our biofilm evolution experiments and because we previously isolated the *fimH* Δ 29-40 in frame mutation and L79 point mutation during evolution of *E. coli* K12 as a gain of function mutation for increased biofilm formation (Yoshida et al 2021). This aspect is now clarified in the legend of Supplementary Fig. S11.

REVIEWERS' COMMENTS:

Reviewer #2 (Remarks to the Author):

The authors have addressed all my comments/concerns. The authors definitively show evolutionary differences between 'biofilm' growth conditions and planktonic growth conditions, and that 'biofilm' bacteria have a reproducible evolutionary trajectory and produce more abx resistant mutants compared to planktonic bacteria. This study, and others that are similar, will help guide clinicians in treatment regimes to minimize resistance development.

Reviewer #3 (Remarks to the Author):

The authors addressed my major concerns about the manuscript through clarification of data presentation. I still believe the manuscripts would be stronger with oscillation experiments to decouple the single tracks of evolution explored in the paper, but appreciated that amount of would may be an undue burden on the authors.